# Offline Reinforcement Learning with Closed-Form Policy Improvement Operators

**Jiachen Li**[1*]  **Edwin Zhang**[1*]  **Ming Yin**[1]  **Qinxun Bai**[2]
**Yu-Xiang Wang**[1]  **William Yang Wang**[1]
[1]UC Santa Barbara    [2] Horizon Robotics
{jiachen_li, ete, ming_yin, yuxiangw, william}@cs.ucsb.edu
qinxun.bai@horizon.ai

## Abstract

Behavior constrained policy optimization has been demonstrated to be a successful paradigm for tackling Offline Reinforcement Learning. By exploiting historical transitions, a policy is trained to maximize a learned value function while constrained by the behavior policy to avoid a significant distributional shift. In this paper, we propose our closed-form policy improvement (CFPI) operators. We make a novel observation that the behavior constraint naturally motivates the use of first-order Taylor approximation, leading to a linear approximation of the policy objective. Additionally, as practical datasets are usually collected by heterogeneous policies, we model the behavior policies as a Gaussian Mixture and overcome the induced optimization difficulties by leveraging the LogSumExp's lower bound and Jensen's Inequality, giving rise to a CFPI operator. We instantiate offline RL algorithms with our novel operators and empirically demonstrate their effectiveness over state-of-the-art algorithms on the standard D4RL benchmark.

## 1  Introduction

Deploying Reinforcement Learning (RL) (Sutton & Barto, 2018) in the real world is hindered by its massive demand for online data. In domains such as robotics (Cabi et al., 2019) and autonomous driving (Sallab et al., 2017), rolling out a premature policy is prohibitively costly and unsafe. To address this issue, offline RL (a.k.a batch RL) (Levine et al., 2020; Lange et al., 2012) has been proposed to learn a policy directly from historical data without environment interaction. However, learning competent policies from a static dataset is challenging. Prior studies have shown that learning a policy without constraining its deviation from the data-generating policies suffers from significant extrapolation errors, leading to training divergence (Fujimoto et al., 2019; Kumar et al., 2019).

Current literature has demonstrated two successful paradigms for managing the trade-off between policy improvement and limiting the distributional shift from the behavior policies. Under the actor-critic framework (Konda & Tsitsiklis, 1999), behavior constrained policy optimization (BCPO) (Fujimoto et al., 2019; Kumar et al., 2019; Fujimoto & Gu, 2021; Wu et al., 2019; Brandfonbrener et al., 2021; Ghasemipour et al., 2021) explicitly regularizes the divergence between learned and behavior policies, while conservative methods (Kumar et al., 2020b; Bai et al., 2022; Yu et al., 2020, 2021) penalize the value estimate for out-of-distribution (OOD) actions to avoid overestimation error. However, most existing model-free offline RL algorithms still require learning off-policy value functions and a target policy through stochastic gradient descent (SGD). Off-policy learning with non-linear function approximators and temporal difference learning (Sutton & Barto, 2018) is notoriously unstable (Kumar et al., 2020a; Mnih et al., 2015; Henderson et al., 2018; Konda & Tsitsiklis, 1999; Watkins & Dayan,

---

[*]Equal Contribution

1992) due to the existence of the deadly-triad (Sutton & Barto, 2018; Van Hasselt et al., 2018). The performance can exhibit significant variance across different random seeds (Islam et al., 2017). In offline settings, learning becomes even more problematic as environment interaction is restricted, thus preventing the learning from corrective feedback (Kumar et al., 2020a). Consequently, training stability poses a major challenge. Although some current approaches (Brandfonbrener et al., 2021) avoids learning an off-policy value function, they still require learning a policy via SGD.

Can we mitigate the issue of learning instability by leveraging optimization techniques? In this paper, we approach this issue from the policy learning perspective, aiming to design a stable policy improvement operator. We take a closer look at the BCPO paradigm and make a novel observation that the requirement of limited distributional shift motivates the use of the first-order Taylor approximation (Callahan, 2010), leading to a linear approximation of the policy objective that is accurate in a sufficiently small neighborhood of the behavior action. Based on this crucial insight, we construct our policy improvement operators that return closed-form solutions by carefully designing a tractable behavior constraint. When modeling the behavior policies as a Single Gaussian, our policy improvement operator deterministically shifts the behavior policy towards a value improving direction derived by solving a Quadratically Constrained Linear Program (QCLP) in closed form. Therefore, our method only requires learning the underlying behavior policies of a given dataset with supervised learning, avoiding the training instability from policy improvement.

Furthermore, we note that practical datasets are likely to be collected by heterogeneous policies, which may give rise to a multimodal behavior action distribution. In this scenario, a Single Gaussian will fail to capture the entire picture of the underlying distribution, limiting the potential of policy improvement. While modeling the behavior as a Gaussian Mixture provides better expressiveness, it incurs extra optimization difficulties due to the non-concavity of its log-likelihood function. We tackle this issue by leveraging the LogSumExp's lower bound and Jensen's inequality, leading to a closed-form policy improvement (CFPI) operator compatible with a multimodal behavior policy. Empirically, we demonstrate the effectiveness of Gaussian Mixture over the conventional Single Gaussian when the underlying distribution comes from hetereogenous policies.

In this paper, we empirically demonstrate that our CFPI operators can instantiate successful offline RL algorithms in a one-step or iterative fashion. Moreover, our methods can also be leveraged to improve a policy learned by the other algorithms. In summary, our main contributions are threefold:

- CFPI operators compatible with single mode and multimodal behavior policies.

- An empirical demonstration of the benefit to model the behavior policy as a Gaussian Mixture in model-free offline RL. To the best of our knowledge, we are the first to do this.

- One-step and iterative instantiations of our algorithm, which outperform state-of-the-art (SOTA) algorithms on the standard D4RL benchmark (Fu et al., 2020).

## 2 Preliminaries

**Reinforcement Learning.** RL aims to maximize returns in a Markov Decision Process (MDP) (Sutton & Barto, 2018) $\mathcal{M} = (\mathcal{S}, \mathcal{A}, R, T, \rho_0, \gamma)$, with state space $\mathcal{S}$, action space $\mathcal{A}$, reward function $R$, transition function $T$, initial state distribution $\rho_0$, and discount factor $\gamma \in [0, 1)$. At each time step $t$, the agent starts from a state $s_t \in \mathcal{S}$, selects an action $a_t \sim \pi(\cdot|s_t)$ from its policy $\pi$, transitions to a new state $s_{t+1} \sim T(\cdot|s_t, a_t)$, and receives reward $r_t := R(s_t, a_t)$. The goal of an RL agent is to learn an optimal policy $\pi^*$ that maximizes the expected discounted cumulative reward $\mathbb{E}_\pi[\sum_{t=0}^\infty \gamma^t r_t]$ without access to the ground truth $R$ and $T$. We define the action value function associated with $\pi$ by $Q^\pi(s, a) = \mathbb{E}_\pi[\sum_{t=0}^\infty \gamma^t r_t | s_0 = s, a_0 = a]$. The RL objective can then be reformulated as

$$\pi^* = \arg\max_\pi J(\pi) := \mathbb{E}_{s \in \rho_0, a \in \pi(\cdot|s)}[Q^\pi(s, a)] \tag{1}$$

In this paper, we consider *offline* RL settings, where we assume restricted access to the MDP $\mathcal{M}$, and a previously collected dataset $\mathcal{D}$ with $N$ transition tuples $\{(s_t^i, a_t^i, r_t^i)\}_{i=1}^N$. We denote the underlying policy that generates $\mathcal{D}$ as $\pi_\beta$, which may or may not be a mixture of individual policies.

**Behavior Constrained Policy Optimization.** One of the critical challenges in offline RL is that the learned $Q$ function tends to assign spuriously high values to OOD actions due to extrapolation error, which is well documented in previous literature (Fujimoto et al., 2019; Kumar et al., 2019).

Behavior Constrained Policy Optimization (BCPO) methods (Fujimoto et al., 2019; Kumar et al., 2019; Fujimoto & Gu, 2021; Wu et al., 2019; Brandfonbrener et al., 2021) explicitly constrain the action selection of the learned policy to stay close to the behavior policy $\pi_\beta$, resulting in a policy improvement step that can be generally summarized by the optimization problem below:

$$\max_\pi \mathbb{E}_{s\sim\mathcal{D}} \left[ \mathbb{E}_{\tilde{a}\sim\pi(\cdot|s)} \left[ Q\left(s, \tilde{a}\right) \right] - \alpha D\left(\pi(\cdot \mid s), \pi_\beta(\cdot \mid s)\right) \right], \qquad (2)$$

where $D(\cdot, \cdot)$ is a divergence function that calculates the divergence between two action distributions, and $\alpha$ is a hyper-parameter controlling the strength of regularization. Consequently, the policy is optimized to maximize the $Q$-value while staying close to the behavior distribution.

Different algorithms may choose different $D(\cdot, \cdot)$ (e.g., KL Divergence (Wu et al., 2019; Jaques et al., 2019), MSE (Fujimoto & Gu, 2021) and MMD (Kumar et al., 2019)). However, to the best of our knowledge, all existing methods tackle this optimization via SGD. In this paper, we take advantage of the regularization and solve the problem in closed form.

## 3 Closed-Form Policy Improvement

In this section, we introduce our policy improvement operators that map the behavior policy to a higher-valued policy, which is accomplished by solving a linearly approximated BCPO. We show that modeling the behavior policy as a Single Gaussian transforms the approximated BCPO into a QCLP and thus can be solved in closed-form (Sec. 3.1). Given that practical datasets are usually collected by heterogeneous policies, we generalize the results by modeling the behavior policies as a Gaussian Mixture to facilitate expressiveness and overcome the incurred optimization difficulties by leveraging the LogSumExp's lower bound (LB) and Jensen's Inequality (Sec. 3.2). We close this section by presenting an offline RL paradigm that leverages our policy improvement operators (Sec. 3.3).

### 3.1 Approximated behavior constrained optimization

We aim to design a learning-free policy improvement operator to avoid learning instability in offline settings. We observe that optimizing towards BCPO's policy objective (2) induces a policy that admits limited deviation from the behavior policy. Consequently, it will only query the $Q$-value within the neighborhood of the behavior action during training, which naturally motivates the employment of the first-order Taylor approximation to derive the following linear approximation of the $Q$ function

$$\begin{aligned} \bar{Q}(s, a; a_\beta) &= (a - a_\beta)^T \left[ \nabla_a Q(s, a) \right]_{a=a_\beta} + Q(s, a_\beta) \\ &= a^T \left[ \nabla_a Q(s, a) \right]_{a=a_\beta} + \text{const.} \end{aligned} \qquad (3)$$

By Taylor's theorem (Callahan, 2010), $\bar{Q}(s, a; a_\beta)$ only provides an accurate linear approximation of $Q(s, a)$ in a sufficiently small neighborhood of $a_\beta$. Therefore, the choice of $a_\beta$ is critical.

Recognizing (2) as a Lagrangian and with the linear approximation (3), we propose to solve the following surrogate problem of (2) given any state $s$:

$$\max_\pi \mathbb{E}_{\tilde{a}\sim\pi} \left[ \tilde{a}^T \left[ \nabla_a Q(s, a) \right]_{a=a_\beta} \right], \quad \text{s.t.} \quad D\left(\pi(\cdot \mid s), \pi_\beta(\cdot \mid s)\right) \le \delta. \qquad (4)$$

Note that it is not necessary for $D(\cdot, \cdot)$ to be a (mathematically defined) divergence measure since any generic $\mathcal{D}(\cdot, \cdot)$ that can constrain the deviation of $\pi$'s action from $\pi_\beta$ can be considered.

**Single Gaussian Behavior Policy.** In general, (4) does not always have a closed-form solution. We analyze a special case where $\pi_\beta = \mathcal{N}(\mu_\beta, \Sigma_\beta)$ is a Gaussian policy, $\pi = \mu$ is a deterministic policy, and $D(\cdot, \cdot)$ is a negative log-likelihood function. In this scenario, a reasonable choice of $\mu$ should concentrate around $\mu_\beta$ to limit distributional shift. Therefore, we set $a_\beta = \mu_\beta$ and the optimization problem (4) becomes the following:

$$\max_\mu \quad \mu^T \left[ \nabla_a Q(s, a) \right]_{a=\mu_\beta}, \quad \text{s.t.} \quad -\log \pi_\beta(\mu|s) \le \delta \qquad (5)$$

We now show that (5) has a closed-form solution and defers the proof the Appendix A.1.

**Proposition 3.1.** *The optimization problem* (5) *has a closed-form solution that is given by*

$$\mu_{sg}(\tau) = \mu_\beta + \frac{\sqrt{2\log\tau}\,\Sigma_\beta \left[ \nabla_a Q(s, a) \right]_{a=\mu_\beta}}{\sqrt{\left[ \nabla_a Q(s, a) \right]_{a=\mu_\beta}^T \Sigma_\beta \left[ \nabla_a Q(s, a) \right]_{a=\mu_\beta}}}, \quad where \;\; \delta = \frac{1}{2}\log\det(2\pi\Sigma_\beta) + \log\tau$$

$$(6)$$

Although we still have to tune $\tau$ as tuning $\alpha$ in (2) for conventional BCPO methods, we have a transparent interpretation of $\tau$'s effect on the action selection thanks to the tractability of (5). Due to the KKT conditions (Boyd et al., 2004), (6) always returns an action $\mu_{\text{sg}}$ with the following property

$$-\log \pi_\beta(\mu|s) = \delta = -\log \frac{1}{\tau} \pi_\beta(\mu_\beta|s) \quad \Longleftrightarrow \quad \pi_\beta(\mu_{\text{sg}}|s) = \frac{1}{\tau} \pi_\beta(\mu_\beta|s) \tag{7}$$

While setting $\tau = 1$ will always return the mean of $\pi_\beta$, a large $\tau$ might send $\mu_{\text{sg}}$ out of the support of $\pi_\beta$, breaking the accuracy guarantee of the first-order Taylor approximation.

### 3.2 Gaussian Mixture as a more expressive model

Performing policy improvement with (6) enjoys favorable computational efficiency and avoids the potential instability caused by SGD. However, its tractability relies on the Single Gaussian assumption of the behavior policy $\pi_\beta$. In practice, the historical datasets are usually collected by heterogeneous policies with different levels of expertise. A Single Gaussian may fail to capture the whole picture of the underlying distribution, motivating the use of a Gaussian Mixture to represent $\pi_\beta$.

$$\pi_\beta = \sum_{i=1}^{N} \lambda_i \mathcal{N}(\mu_i, \Sigma_i), \quad \sum_{i=1}^{N} \lambda_i = 1 \tag{8}$$

However, directly plugging the Gaussian Mixture $\pi_\beta$ into (5) breaks its tractability, resulting in a non-convex optimization below

$$\max_\mu \quad \mu^T \left[\nabla_a Q(s,a)\right]_{a=a_\beta},$$
$$\text{s.t.} \quad \log \sum_{i=1}^{N} \left(\lambda_i \det(2\pi\Sigma_i)^{-\frac{1}{2}} \exp\left(-\frac{1}{2}(\mu - \mu_i)^T \Sigma_i^{-1}(\mu - \mu_i)\right)\right) \geq -\delta \tag{9}$$

We are confronted with two major challenges to solve the optimization problem (9). First, it is unclear how to choose a proper $a_\beta$ while we need to ensure that the solution $\mu$ lies within a small neighborhood of $a_\beta$. Second, the constraint of (9) does not admit a convex form, posing non-trivial optimization difficulties. We leverage the lemma below to tackle the non-convexity of the constraint.

**Lemma 3.1.** $\log \sum_{i=1}^{N} \lambda_i \exp(x_i)$ *admits the following inequality:*

1. *(LogSumExp's LB)* $\quad \log \sum_{i=1}^{N} \lambda_i \exp(x_i) \geq \max_i \{x_i + \log \lambda_i\}$

2. *(Jensen's Inequality)* $\quad \log \sum_{i=1}^{N} \lambda_i \exp(x_i) \geq \sum_{i=1}^{N} \lambda_i x_i$

Next, we show that applying each inequality in Lemma 3.1 to the constraint of (9) respectively resolves the intractability and leads to natural choices of $a_\beta$.

**Proposition 3.2.** *By applying the first inequality of Lemma 3.1 to the constraint of* (9)*, we can derive an optimization problem that lower bounds* (9)

$$\max_\mu \quad \mu^T \left[\nabla_a Q(s,a)\right]_{a=a_\beta},$$
$$\text{s.t.} \quad \max_i \left\{-\frac{1}{2}(\mu - \mu_i)^T \Sigma_i^{-1}(\mu - \mu_i) - \frac{1}{2}\log\det(2\pi\Sigma_i) + \log\lambda_i\right\} \geq -\delta, \tag{10}$$

*and the closed-form solution to* (10) *is given by*

$$\mu_{lse}(\tau) = \arg\max_{\bar{\mu}_i(\delta)} \bar{\mu}_i^T \left[\nabla_a Q(s,a)\right]_{a=\mu_i}, \quad \text{s.t.} \quad \delta = \min_i\{\frac{1}{2}\log\det(2\pi\Sigma_i) - \log\lambda_i\} + \log\tau$$
$$\text{where} \quad \bar{\mu}_i(\delta) = \mu_i + \sqrt{\frac{2(\delta + \log\lambda_i) - \log\det(2\pi\Sigma_i)}{\left[\nabla_a Q(s,a)\right]_{a=\mu_i}^T \Sigma_i \left[\nabla_a Q(s,a)\right]_{a=\mu_i}}} \Sigma_i \left[\nabla_a Q(s,a)\right]_{a=\mu_i} \tag{11}$$

**Proposition 3.3.** *By applying the second inequality of Lemma 3.1 to the constraint of* (9)*, we can derive an optimization problem that lower bounds* (9)

$$\max_\mu \mu^T \left[\nabla_a Q(s,a)\right]_{a=a_\beta}, \quad \text{s.t.} \quad \sum_{i=1}^{N} \lambda_i \left(-\frac{1}{2}\log\det(2\pi\Sigma_i) - \frac{1}{2}(\mu - \mu_i)^T \Sigma_i^{-1}(\mu - \mu_i)\right) \geq -\delta \tag{12}$$

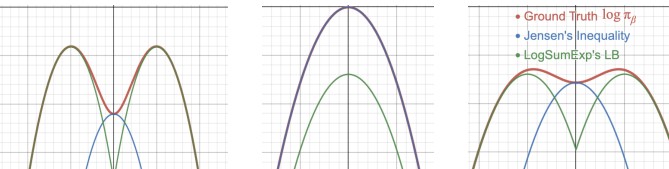

Figure 1: Apply Lemma 3.1 to Gaussian Mixture's log probability $\log \pi_\beta$ at different scenarios. (L) $\log \pi_\beta$ has multiple modes. LogSumExp's LB preserves multimodality. (M) $\log \pi_\beta$ reduces to Single Gaussian. Jensen's inequality becomes equality. (R) $\log \pi_\beta$ is similar to a uniform distribution.

*and the closed-form solution to* (12) *is given by*

$$
\mu_{jensen}(\tau) = \overline{\mu} + \sqrt{\frac{2 \log \tau - \sum_{i=1}^{N} \lambda_i \mu_i^T \Sigma_i^{-1} \mu_i + \overline{\mu}^T \overline{\Sigma}^{-1} \overline{\mu}}{[\nabla_a Q(s,a)]_{a=\overline{\mu}}^T \overline{\Sigma} [\nabla_a Q(s,a)]_{a=\overline{\mu}}}} \; \overline{\Sigma} [\nabla_a Q(s,a)]_{a=\overline{\mu}},
$$

$$
where \; \overline{\Sigma} = \left( \sum_{i=1}^{N} \lambda_i \Sigma_i^{-1} \right)^{-1}, \quad \overline{\mu} = \overline{\Sigma} \left( \sum_{i=1}^{N} \lambda_i \Sigma_i^{-1} \mu_i \right), \quad \delta = \log \tau + \frac{1}{2} \sum_{i=1}^{N} \lambda_i \log \det(2\pi\Sigma_i)
$$

(13)

We defer the detailed proof of Proposition 3.2 and Proposition 3.3 as well as how we choose $a_\beta$ for each optimization problem to Appendix A.2 and A.3, respectively.

Indeed, these two optimization problems have their own assets and liabilities. When $\pi_\beta$ exhibits an obvious multimodality as is shown in Fig. 1 (L), the lower bound of $\log \pi_\beta$ constructed by Jensen's Inequality cannot capture different modes due to its concavity, losing the advantage of modeling $\pi_\beta$ as a Gaussian Mixture. In this case, the optimization problem (10) can serve as a reasonable surrogate problem of (9), as LogSumExp's LB still preserves the multimodality of $\log \pi_\beta$.

When $\pi_\beta$ is reduced to a Single Gaussian, the approximation with the Jensen's Inequality becomes equality as is shown in Fig. 1 (M). Thus $\mu_{\text{jensen}}$ returned by (13) exactly solves the optimization problem (9). However, in this case, the tightness of LogSumExp's LB largely depends on the weights $\lambda_{i=1...N}$. If each Gaussian component is distributed and weighted identically, the lower bound will be $\log N$ lower than the actual value. Moreover, there also exists the scenario (Fig. 1 (R)) when both (10) and (12) can serve as reasonable surrogates to the original problem (9).

Fortunately, we can combine the best of both worlds and design a policy improvement operator accounting for all the above scenarios. As both Proposition 3.2 and 3.3 have closed-form solutions, the operator returns a policy that selects the higher-valued action from $\mu_{\text{lse}}$ and $\mu_{\text{jensen}}$

$$
\mu_{\text{mg}}(\tau) = \arg\max_\mu Q(s,\mu), \quad \text{s.t.} \quad \mu \in \{\mu_{\text{lse}}(\tau), \mu_{\text{jensen}}(\tau)\}
$$

(14)

### 3.3 Algorithm template

We have derived two CFPI operators that map the behavior policy to a higher-valued policy. When the behavior policy $\pi_\beta$ is a Single Gaussian, $\mathcal{I}_{\text{SG}}(\pi_\beta, Q; \tau)$ returns a policy with action selected by (6). When $\pi_\beta$ is a Gaussian Mixture, $\mathcal{I}_{\text{MG}}(\pi_\beta, Q; \tau)$ returns a policy with action selected by (14). We note that our methods can also work with a non-Gaussian $\pi_\beta$. Appendix D provides the derivations for the corresponding CFPI operators when $\pi_\beta$ is modeled as both a deterministic policy and VAE. Algorithm 1 shows that our CFPI operators enable the design of a general offline RL template that can yield one-step, multi-step and iterative methods, where $\mathcal{E}$ is a general policy evaluation operator that returns a value function $\hat{Q}_t$. When setting $T = 0$, we obtain our one-step method. We defer the discussion on multi-step and iterative methods to the Appendix C.

While the design of our CFPI operators is motivated from the behavior constraint, we highlight that they are compatible with general baseline policies $\pi_b$ besides $\pi_\beta$. Sec. 5.2 and Appendix G.7 show that our CFPI operators can improve policies learned by IQL and CQL (Kumar et al., 2020b).

### 3.4 Theoretical guarantees for closed-form policy improvement

At a high level, Algorithm 1 follows the *approximate policy iteration* (API) (Perkins & Precup, 2002) by iterating over the policy evaluation ($\mathcal{E}$ step, Line 4) and policy improvement ($\mathcal{I}$ step, Line

---

**Algorithm 1** Offline RL with closed-form policy improvement operators (CFPI)

---

**Input**: Dataset $\mathcal{D}$, baseline policy $\hat{\pi}_b$, value function $\hat{Q}_{-1}$, HP $\tau$

1: Warm start $\hat{Q}_0 = \text{SARSA}(\hat{Q}_{-1}, \mathcal{D})$ with the SARSA-style algorithm (Sutton & Barto, 2018)
2: Get one-step policy $\hat{\pi}_1 = \mathcal{I}(\hat{\pi}_b, \hat{Q}_0; \tau)$
3: **for** $t = 1 \ldots T$ **do**
4:     Policy evaluation: $\hat{Q}_t = \mathcal{E}(\hat{Q}_{t-1}, \hat{\pi}_t, \mathcal{D})$
5:     Get policy: $\hat{\pi}_{t+1} = \mathcal{I}(\hat{\pi}_b, \hat{Q}_t; \tau)$     (concrete choices of $\mathcal{I}$ includes $\mathcal{I}_{\text{MG}}$ and $\mathcal{I}_{\text{SG}}$)
6: **end for**

---

5). Therefore, to verify $\mathcal{E}$ provides the improvement, we need to first show policy evaluation $\hat{Q}_t$ is accurate. We employ the Fitted Q-Iteration (Sutton & Barto, 2018) to perform policy evaluation, which is known to be statistically efficient (*e.g.* (Chen & Jiang, 2019)) under the mild condition for the function approximation class. Next, for the performance gap between $J(\hat{\pi}_{t+1}) - J(\hat{\pi}_t)$, we apply the standard performance difference lemma (Kakade & Langford, 2002; Kakade, 2003).

**Theorem 3.1.** *[Safe Policy Improvement] Assume the state and action spaces are discrete.[2] Let $\hat{\pi}_1$ be the policy obtained after the CFPI update (Line 2 of Algorithm 1). Then with probability $1 - \delta$,*

$$J(\hat{\pi}_1) - J(\hat{\pi}_\beta) \geq \frac{1}{1-\gamma} \mathbb{E}_{s \sim d^{\hat{\pi}_1}} \left[ \bar{Q}^{\hat{\pi}_\beta}(s, \hat{\pi}_1(s)) - \bar{Q}^{\hat{\pi}_\beta}(s, \hat{\pi}_\beta(s)) \right]$$

$$- \frac{2}{1-\gamma} \mathbb{E}_{s \sim d^{\hat{\pi}_1}} \mathbb{E}_{a \sim \hat{\pi}_1(\cdot|s)} \left[ \frac{C_{\gamma,\delta}}{\sqrt{\mathcal{D}(s,a)}} + C_{\text{CFPI}}(s,a) \right] := \zeta.$$

*For multi-step $T$ iterative update, we similarly have with probability $1 - \delta$,*

$$J(\hat{\pi}_T) - J(\hat{\pi}_\beta) = \sum_{t=1}^{T} J(\hat{\pi}_t) - J(\hat{\pi}_{t-1}) \geq \sum_{t=1}^{T} \zeta^{(t)},$$

*where $\mathcal{D}(s,a)$ denotes number of samples at $s,a$, $C_{\gamma,\delta}$ denotes the learning coefficient of SARSA and $C_{\text{CFPI}}(s,a)$ denotes the first-order approximation error from* (3). *We defer detailed derivation and the expression of $C_{\gamma,\delta}, \zeta^{(t)}$ and $C_{\text{CFPI}}(s,a)$ in Appendix A.4. When $a = a_\beta$, $C_{\text{CFPI}}(s,a) = 0$.*

By Theorem 3.1, $\hat{\pi}_1$ is a $\zeta$-safe improved policy. The $\zeta$ safeness consists of two parts: $C_{\text{CFPI}}$ is caused by the first-order approximation, and the $C_{\gamma,\delta}/\sqrt{\mathcal{D}(s,a)}$ term is incurred by the SARSA update. Similarly, $\hat{\pi}_T$ is a $\sum_{t=1}^{T} \zeta^{(t)}$-safe improved policy.

# 4 Related Work

Our methods belong and are motivated by the successful BCPO paradigm, which imposes constraints as in (2) to prevent from selecting OOD actions. Algorithms from this paradigm may apply different divergence functions, e.g., KL-divergence (Wu et al., 2019; Jaques et al., 2019), MMD (Kumar et al., 2019) or the MSE (Fujimoto & Gu, 2021). All these methods perform policy improvement via SGD. Instead, we perform CFPI by solving a linear approximation of (2). Another line of research enforces the behavior constraint via parameterization. BCQ (Fujimoto et al., 2019) learns a generative model as the behavior policy and a $Q$ function to select the action from a set of perturbed behavior actions. Ghasemipour et al. (2021) further show that the perturbation model can be discarded.

The design of our CFPI operators is inspired by the SOTA online RL algorithm OAC (Ciosek et al., 2019). It treats the evaluation policy as the baseline $\pi_b$ and obtains an optimistic exploration policy by solving a similar optimization problem as (5). We extend the result to accommodate a multi-modal $\pi_b$ and overcome the optimization difficulties by leveraging Lemma 3.1. In Appendix H, we further draw connections with prior works that leveraged the Taylor expansion approach to RL.

Recently, one-step (Kostrikov et al., 2021; Brandfonbrener et al., 2021) algorithms have achieved great success. Instead of iteratively performing policy improvement and evaluation, these methods

---

[2]Note here we assume the discreteness only for the purpose of analysis. For the more general cases, please refer to Appendix A.4. In Theorem 3.1, $\mu_h^\pi(s, a|s_0, a_0) := P^\pi(s_h = s, a_h = a, |s_0 = s, a_0 = a)$.

Table 1: Comparison between our one-step policy and SOTA methods on the Gym-MuJoCo domain of D4RL. Our method uses the same $\tau$ for all datasets except Hopper-M-E (detailed in Appendix F.1). We report the mean and standard deviation of our method's performance across 10 seeds. Each seed contains an individual training process and evaluates the policy for 100 episodes. We use Cheetah for HalfCheetah, M for Medium, E for Expert, and R for Replay. We bold the best results for each task.

| Dataset | SG-BC | MG-BC | DT | TT | OnestepRL | TD3+BC | CQL | IQL | Our $\mathcal{I}_{\text{MG}}(\hat{\pi}_\beta, \hat{Q}_0)$ |
|---|---|---|---|---|---|---|---|---|---|
| Cheetah-M-v2 | 40.6 | 40.6 | 42.6 | 46.9 | **55.6** | 48.3 | 44.0 | 47.4 | $52.1 \pm 0.3$ |
| Hopper-M-v2 | 53.7 | 53.9 | 67.6 | 61.1 | 83.3 | 59.3 | 58.5 | 66.2 | $\mathbf{86.8 \pm 4.0}$ |
| Walker2d-M-v2 | 71.9 | 70.0 | 74.0 | 79.8 | 85.6 | 83.7 | 72.5 | 78.3 | $\mathbf{88.3 \pm 1.6}$ |
| Cheetah-M-R-v2 | 34.9 | 33.0 | 36.6 | 41.9 | 42.4 | 44.6 | **45.5** | 44.2 | $44.5 \pm 0.4$ |
| Hopper-M-R-v2 | 12.4 | 21.2 | 82.7 | 91.5 | 71.0 | 60.9 | **95.0** | 94.7 | $93.6 \pm 7.9$ |
| Walker2d-M-R-v2 | 22.9 | 22.8 | 66.6 | **82.6** | 71.6 | 81.8 | 77.2 | 73.8 | $78.2 \pm 5.6$ |
| Cheetah-M-E-v2 | 46.6 | 51.7 | 86.8 | 95.0 | 93.5 | 90.7 | 91.6 | 86.7 | $\mathbf{97.3 \pm 1.8}$ |
| Hopper-M-E-v2 | 53.9 | 69.2 | **107.6** | 101.9 | 102.1 | 98.0 | 105.4 | 91.5 | $104.2 \pm 5.1$ |
| Walker2d-M-E-v2 | 92.3 | 93.2 | 108.1 | 110.0 | **110.9** | 110.1 | 108.8 | 109.6 | $\mathbf{111.9 \pm 0.3}$ |
| Total | 429.1 | 455.6 | 672.6 | 710.1 | 716.0 | 677.4 | 698.5 | 692.4 | $\mathbf{757.0 \pm 27.0}$ |

Table 2: Comparison between our Iterative $\mathcal{I}_{\text{MG}}$ and SOTA methods on the AntMaze domain. We report the mean and standard deviation across 5 seeds for our method with each seed evaluating for 100 episodes. The performance for all baselines is directly reported from the IQL paper. Our Iterative $\mathcal{I}_{\text{MG}}$ outperforms all baselines on 5 out of 6 tasks and obtains the best overall performance.

| Dataset | BC | DT | Onestep RL | TD3+BC | CQL | IQL | Iterative $\mathcal{I}_{\text{MG}}$ |
|---|---|---|---|---|---|---|---|
| antmaze-u-v0 | 54.6 | 59.2 | 64.3 | 78.6 | 74.0 | 87.5 | $\mathbf{90.2 \pm 3.9}$ |
| antmaze-u-d-v0 | 45.6 | 49.3 | 60.7 | 71.4 | **84.0** | 62.2 | $58.6 \pm 15.2$ |
| antmaze-m-p-v0 | 0.0 | 0.0 | 0.3 | 10.6 | 61.2 | 71.2 | $\mathbf{75.2 \pm 6.9}$ |
| antmaze-m-d-v0 | 0.0 | 0.7 | 0.0 | 3.0 | 53.7 | 70.0 | $\mathbf{72.2 \pm 7.3}$ |
| antmaze-l-p-v0 | 0.0 | 0.0 | 0.0 | 0.2 | 15.8 | 39.6 | $\mathbf{51.4 \pm 7.7}$ |
| antmaze-l-d-v0 | 0.0 | 1.0 | 0.0 | 0.0 | 14.9 | 47.5 | $\mathbf{52.4 \pm 10.9}$ |
| Total | 100.2 | 112.2 | 125.3 | 163.8 | 303.6 | 378.0 | $\mathbf{400.0 \pm 52.0}$ |

only learn a $Q$ function via SARSA without bootstrapping from OOD action value. These methods further apply an policy improvement operator (Wu et al., 2019; Peng et al., 2019) to extract a policy. We also instantiate a one-step algorithm with our CFPI operator and evaluate on standard benchmarks.

## 5 Experiments

Our experiments aim to demonstrate the effectiveness of our CFPI operators. Firstly, on the standard offline RL benchmark D4RL (Fu et al., 2020), we show that instantiating offline RL algorithms with our CFPI operators in both one-step and iterative manners outperforms SOTA methods (Sec. 5.1). Secondly, we show that our CFPI operator can improve a policy learned by other algorithms (Sec. 5.2). Ablation studies in Sec. 5.3 further shows our superiority over the other policy improvement operators and demonstrate the benefit of modeling the behavior policy as a Gaussian Mixture.

### 5.1 Comparison with SOTA offline RL algorithms

We instantiate a one-step offline RL algorithm from Algorithm 1 with our policy improvement operator $\mathcal{I}_{\text{MG}}$. We learned a Gaussian Mixture baseline policy $\hat{\pi}_\beta$ via behavior cloning. We employed the IQN (Dabney et al., 2018a) architecture to model the Q value network for its better generalizability, as we need to estimate out-of-buffer $Q(s, a)$ during policy deployment. We trained the $\hat{Q}_0$ with SARSA algorithm (Sutton & Barto, 2018; Parisotto et al., 2015). Appendix F.1 includes detailed training procedures of $\hat{\pi}_\beta$ and $\hat{Q}_0$ with full HP settings. We obtain our one-step policy as $\mathcal{I}_{\text{MG}}(\hat{\pi}_\beta, \hat{Q}_0; \tau)$.

We evaluate our one-step algorithm on the D4RL benchmark focusing on the Gym-MuJoCo domain, which contains locomotion tasks with dense rewards. Table 1 compares our one-step algorithm with SOTA methods, including the other one-step actor-critic methods IQL (Kostrikov et al., 2021), OneStepRL (Brandfonbrener et al., 2021), BCPO method TD3+BC (Fujimoto & Gu, 2021), conservative method CQL (Kumar et al., 2020b), and trajectory optimization methods DT (Chen et al.,

2021), TT (Janner et al., 2021). We also include the performance of two behavior policies SG-BC and MG-BC modeled with Single Gaussian and Gaussian Mixture, respectively. We directly report results for IQL, BCQ, TD3+BC, CQL, and DT from the IQL paper, and TT's result from its own paper. Note that OneStepRL instantiates three different algorithms. We only report its (Rev. KL Reg) result because this algorithm follows the BCPO paradigm and achieves the best overall performance. We highlight that OnesteRL reports the results by tuning the HP for each dataset.

Results in Table 1 demonstrate that our one-step algorithm outperforms the other algorithms by a significant margin without training a policy to maximize its $Q$-value through SGD. We note that we use the same $\tau$ for all datasets except Hopper-M-E. In Sec. 5.3, we will perform ablation studies and provide a fair comparison between our CFPI operators and the other policy improvement operators.

We further instantiate an iterative algorithm with $\mathcal{I}_{MG}$ and evaluate its effectiveness on the challenging AntMaze domain of D4RL. The 6 tasks from AntMaze are more challenging due to their sparse-reward nature and lack of optimal trajectories in the static datasets. Table 2 compares our Iterative $\mathcal{I}_{MG}$ with SOTA algorithms on the AntMaze domain. Our method uses the same set of HP for all 6 tasks, outperforming all baselines on 5 out of 6 tasks and obtaining the best overall performance. Appendix C.1 presents additional details with pseudo-codes and training curves.

## 5.2 Improvement over a learned policy

In this section, we show that our CFPI operator $\mathcal{I}_{SG}$ can further improve the performance of a Single Gaussian policy $\pi_{IQL}$ learned by IQL (Kostrikov et al., 2021) on the AntMaze domain. We first obtain the IQL policy $\pi_{IQL}$ and $Q_{IQL}$ by training for 1M gradient steps using the PyTorch Implementation from RLkit (Berkeley). We emphasize that we follow the authors' exact training and evaluation protocols and include all training curves in Appendix G.6. Interestingly, while the running average of the evaluation results during the course of training matches the reported results in the IQL paper, Table 3 shows that the evaluation of the final 1M-step policy $\pi_{IQL}$ does not

Table 3: Our $\mathcal{I}_{SG}(\pi_{IQL}, Q_{IQL})$ improves over the policy $\pi_{IQL}$ learned by IQL on AntMaze. We report the mean and standard deviation 10 seeds. Each seed evaluates for 100 episodes.

| Dataset | $\pi_{IQL}$ (train) | $\pi_{IQL}$ (1M) | $\mathcal{I}_{SG}(\pi_{IQL}, Q_{IQL})$ |
|---|---|---|---|
| antmaze-u-v0 | **87.4 ± 3.2** | 83.6 ± 3.2 | 85.1 ± 5.3 |
| antmaze-u-d-v0 | **59.0 ± 5.7** | 55.8 ± 7.9 | 55.0 ± 9.1 |
| antmaze-m-p-v0 | 71.1 ± 5.43 | 64.2 ± 13.2 | **75.5 ± 6.1** |
| antmaze-m-d-v0 | 70.0 ± 6.16 | 66.8 ± 9.4 | **79.9 ± 3.8** |
| antmaze-l-p-v0 | 34.4 ± 6.04 | 35.6 ± 7.0 | **37.7 ± 7.7** |
| antmaze-l-d-v0 | 39.8 ± 9.09 | 38.8 ± 7.1 | **40.1 ± 5.6** |
| Total | 361.7 ± 35.6 | 344.7 ± 47.8 | **373.3 ± 37.5** |

match the reported performance on all 6 tasks, **echoing the training instability we are trying to resolve with our CFPI operators**. This demonstrates how drastically performance can fluctuate across just dozens of epochs. Thanks to the tractability of $\mathcal{I}_{SG}$, we directly obtain an improved policy $\mathcal{I}_{SG}(\pi_{IQL}, Q_{IQL}; \tau)$ that achieves better overall performance than both $\pi_{IQL}$ (train) and (1M), as shown in Table 3. We tune the HP $\tau$ using a small set of seeds for each task following the practice of (Brandfonbrener et al., 2021; Fu et al., 2020) and include more details in Appendix F.2 and G.6.

## 5.3 Ablation studies

We first provide a fair comparison with the other policy improvement operators, demonstrating the effectiveness of solving the approximated BCPO (4) and modeling the behavior policy as a Gaussian Mixture. Additionally, we examine the sensitivity on $\tau$, ablate the number of Gaussian components, and discuss the limitation by ablating the $Q$ network in Appendix G.2, G.3, G.4, respectively.

**Effectiveness of our CFPI operators.** In Table 4, we compare our CFPI operators with two policy improvement operators, namely, Easy BCQ (EBCQ) and Rev. KL Reg from OneStepRL (Brandfonbrener et al., 2021). EBCQ doe not require training either, returning a policy by selecting an action that maximizes a learned $\hat{Q}$ from $N_{bcq}$ actions randomly sampled from the behavior policy $\hat{\pi}_\beta$. Rev. KL Reg sets $D(\cdot, \cdot)$ in (2) as the reverse KL divergence and solves the problem via SGD, with $\alpha$ controlling the regularization strength. We omit the comparison with the other learning-based operator Exp. Weight, as Rev. KL Reg achieves the best overall performance in OneStepRL.

For all methods, we present results with $\hat{\pi}_\beta$ modeled by Single Gaussian (SG-) and Gaussian Mixture (MG-). To ensure a fair comparison, we employ the same $\hat{Q}_0$ and $\hat{\pi}_\beta$ modeled and learned in the same way as in Sec. 5.1 for all methods. Moreover, we tune $N_{bcq}$ for EBCQ, $\alpha$ for Rev. KL Reg, and

Table 4: Ablation studies of our Method on the Gym-MuJoCo domain. Again we report the mean and std of 10 seeds, each seed evaluates for 100 episodes.

| Dataset | SG-EBCQ | MG-EBCQ | SG-Rev. KL Reg | MG-Rev. KL Reg | $\mathcal{I}_{\mathrm{SG}}$ | $\mathcal{I}_{\mathrm{MG}}$ |
|---|---|---|---|---|---|---|
| Cheetah-M-v2 | $\mathbf{53.3 \pm 0.2}$ | $51.5 \pm 0.2$ | $47.1 \pm 0.2$ | $47.0 \pm 0.2$ | $51.1 \pm 0.1$ | $52.1 \pm 0.3$ |
| Hopper-M-v2 | $\mathbf{86.8 \pm 5.2}$ | $82.5 \pm 1.9$ | $70.3 \pm 7.0$ | $76.3 \pm 6.9$ | $75.6 \pm 3.7$ | $\mathbf{86.8 \pm 4.0}$ |
| Walker2d-M-v2 | $85.2 \pm 5.1$ | $85.2 \pm 2.1$ | $82.4 \pm 1.0$ | $82.8 \pm 1.8$ | $88.1 \pm 1.1$ | $\mathbf{88.3 \pm 1.6}$ |
| Cheetah-M-R-v2 | $43.5 \pm 0.6$ | $43.0 \pm 0.3$ | $44.3 \pm 0.4$ | $44.4 \pm 0.5$ | $42.8 \pm 0.4$ | $\mathbf{44.5 \pm 0.4}$ |
| Hopper-M-R-v2 | $88.5 \pm 12.2$ | $83.6 \pm 10.3$ | $\mathbf{99.7 \pm 1.0}$ | $99.4 \pm 2.1$ | $87.7 \pm 8.7$ | $93.6 \pm 7.9$ |
| Walker2d-M-R-v2 | $75.4 \pm 4.6$ | $73.1 \pm 5.2$ | $63.6 \pm 28.5$ | $69.7 \pm 30.9$ | $71.3 \pm 4.4$ | $\mathbf{78.2 \pm 5.6}$ |
| Cheetah-M-E-v2 | $81.8 \pm 5.4$ | $84.5 \pm 4.6$ | $78.9 \pm 9.8$ | $65.0 \pm 10.1$ | $91.1 \pm 3.1$ | $\mathbf{97.3 \pm 1.8}$ |
| Hopper-M-E-v2 | $40.0 \pm 5.8$ | $56.1 \pm 6.2$ | $76.6 \pm 18.3$ | $\mathbf{79.4 \pm 32.6}$ | $70.3 \pm 8.9$ | $73.0 \pm 10.5$ |
| Walker2d-M-E-v2 | $111.1 \pm 1.8$ | $111.1 \pm 1.0$ | $106.7 \pm 4.1$ | $107.1 \pm 4.0$ | $111.1 \pm 1.1$ | $\mathbf{111.9 \pm 0.3}$ |
| Total | $665.5 \pm 41.0$ | $670.6 \pm 31.9$ | $669.7 \pm 70.3$ | $671.2 \pm 89.1$ | $688.9 \pm 31.6$ | $\mathbf{725.8 \pm 32.4}$ |

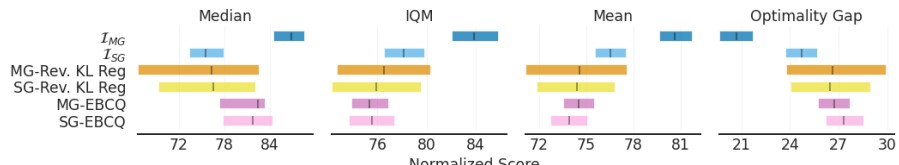

Figure 2: Aggregate metrics (Agarwal et al., 2021) with 95% CIs based on results reported in Table 4. The CIs are estimated using the percentile bootstrap with stratified sampling. Higher median, IQM, and mean scores, and lower Optimality Gap correspond to better performance. Our $\mathcal{I}_{\mathrm{MG}}$ outperforms baselines by a significant margin based on all four metrics. Appendix E includes additional details.

$\tau$ for our methods. Each method uses the same set of HP for all datasets. As a result, the Hopper-M-E performance of $\mathcal{I}_{\mathrm{MG}}$ reported in Table 4 is different from Table 1. Appendix F.1 includes further details on the HP tuning and corresponding experiment results in Table 9, 10, 11 and 12.

As is shown in Table 4 and Fig. 2, our $\mathcal{I}_{\mathrm{MG}}$ clearly outperforms all baselines by a significant margin. The learning-based method Rev. KL Reg exhibits a substantial amount of variance, again echoing the training instability we are trying to resolve. Moreover, our CFPI operators outperform their EBCQ counterparts, demonstrating the effectiveness of solving the approximated BCPO.

**Effectiveness of Gaussian Mixture.** As the three M-E datasets are collected by an expert and medium policy, we should recover an expert performance as long as we can 1) capture the two modes of the action distribution 2) and always select action from the expert mode. In other words, we can leverage the $\hat{Q}_0$ learned by SARSA to select actions from the mean of each Gaussian component, resulting in a mode selection algorithm (MG-MS) that selects its action by

$$\mu_{\mathrm{mode}} = \arg\max_{\hat{\mu}_i} \quad \hat{Q}_0(s, \hat{\mu}_i), \quad \text{s.t.} \quad \{\hat{\mu}_i | \hat{\lambda}_i > \xi\}, \quad \text{where} \quad \sum_{i=1:N} \hat{\lambda}_i \mathcal{N}(\hat{\mu}_i, \hat{\Sigma}_i) = \hat{\pi}_\beta, \quad (15)$$

$\xi$ is set to filter out trivial components. Our MG-MS achieves an expert performance on Hopper-M-E $(104.2 \pm 5.1)$ and Walker2d-M-E $(104.1 \pm 6.7)$, and matches SOTA algorithms in Cheetah-M-E $(91.3 \pm 2.1)$. Appendix G.1 includes the full results of MG-MS on the Gym MuJoCo domain.

## 6 Conclusion and Limitations

Motivated by the behavior constraint in the BCPO paradigm, we propose CFPI operators that perform policy improvement by solving an approximated BCPO in closed form. As practical datasets are usually generated by heterogeneous policies, we use the Gaussian Mixture to model the data-generating policies and overcome extra optimization difficulties by leveraging the LogSumExp's LB and Jensen's Inequality. We instantiate a one-step offline RL algorithm with our CFPI operator and show that it can outperform SOTA algorithms on the Gym-MuJoCo domain of the D4RL benchmark.

Our CFPI operators avoid the training instability incurred by policy improvement through SGD. However, our method still requires learning a good $Q$ function. Specifically, our operators rely on the gradient information provided by the $Q$, and its accuracy largely impacts the effectiveness of our policy improvement. Therefore, one promising future direction for this work is to investigate ways to robustify the policy improvement given a noisy $Q$.

## Reproducibility Statement

We include our codes in the supplementary material.

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

# Appendix

**Outline of the Appendix**

In this Appendix, we organize the content in the following ways:

- Appendix A presents the missing proofs for Proposition 3.1, 3.2, 3.3 and Theorem 3.1 in the main paper.
- Appendix B justify one HP setting for Equation 14.
- Appendix C discusses how to instantiate multi-step and iterative algorithms from our algorithm template Algorithm 1.
- Appendix D provides the derivation of a new CFPI operator that can work with both deterministic and VAE policy.
- Appendix E conducted a reliable evaluation to demonstrate the statistical significance of our methods and address statistical uncertainty.
- Appendix F gives experiment details, HP settings and corresponding experiment results.
- Appendix G provides additional ablation studies and experiment results.
- Appendix H includes additional related work and discusses the relationship between our method and prior literature that leverage the Taylor expansion approach.

Our experiments are conducted on various types of 8GPUs machines. Different machines may have different GPU types, such as NVIDIA GA100 and TU102. Training a behavior policy for 500K gradient steps takes around 40 minutes, while training a $Q$ network for 500K gradient steps takes around 50 minutes.

# A  Proofs and Theoretical Results

## A.1  Proof of Proposition 3.1

**Proposition 3.1.** *The optimization problem* (5) *has a closed-form solution that is given by*

$$\mu_{sg}(\tau) = \mu_\beta + \frac{\sqrt{2\log\tau}\,\Sigma_\beta\,[\nabla_a Q(s,a)]_{a=\mu_\beta}}{\sqrt{[\nabla_a Q(s,a)]_{a=\mu_\beta}^T\,\Sigma_\beta\,[\nabla_a Q(s,a)]_{a=\mu_\beta}}}, \quad where \;\; \delta = \frac{1}{2}\log\det(2\pi\Sigma_\beta) + \log\tau$$

$$\tag{16}$$

*Proof.* The optimization problem (5) can be converted into the QCLP

$$\max_\mu \quad \mu^T\,[\nabla_a Q(s,a)]_{a=\mu_\beta}, \quad \text{s.t.} \quad \frac{1}{2}(\mu-\mu_\beta)^T\Sigma_\beta^{-1}(\mu-\mu_\beta) \le \delta - \frac{1}{2}\log\det(2\pi\Sigma_\beta) \tag{17}$$

Following a similar procedure as is in OAC (Ciosek et al., 2019), we first derive the Lagrangian below:

$$L = \mu^T\,[\nabla_a Q(s,a)]_{a=\mu_\beta} - \eta\left(\frac{1}{2}(\mu-\mu_\beta)^T\Sigma_\beta^{-1}(\mu-\mu_\beta) - \delta + \frac{1}{2}\log\det(2\pi\Sigma_\beta)\right) \tag{18}$$

Taking the derivatives w.r.t $\mu$, we get

$$\nabla_\mu L = [\nabla_a Q(s,a)]_{a=\mu_\beta} - \eta\Sigma_\beta^{-1}(\mu-\mu_\beta) \tag{19}$$

By setting $\nabla_\mu L = 0$, we get

$$\mu = \mu_\beta + \frac{1}{\eta}\Sigma_\beta\,[\nabla_a Q(s,a)]_{a=\mu_\beta} \tag{20}$$

To satisfy the the KKT conditions (Boyd et al., 2004), we have $\eta > 0$ and

$$(\mu-\mu_\beta)^T\Sigma_\beta^{-1}(\mu-\mu_\beta) = 2\delta - \log\det(2\pi\Sigma_\beta) \tag{21}$$

Finally with (20) and (21), we get

$$\eta = \sqrt{\frac{[\nabla_a Q(s,a)]_{a=\mu_\beta}^T\,\Sigma_\beta\,[\nabla_a Q(s,a)]_{a=\mu_\beta}}{2\delta - \log\det(2\pi\Sigma_\beta)}} \tag{22}$$

By setting $\delta = \frac{1}{2}\log\det(2\pi\Sigma_\beta) + \log\tau$ and plugging (22) to (20), we obtain the final solution as

$$\mu_{\text{sg}}(\tau) = \mu_\beta + \frac{\sqrt{2\log\tau}\,\Sigma_\beta\,[\nabla_a Q(s,a)]_{a=\mu_\beta}}{\sqrt{[\nabla_a Q(s,a)]_{a=\mu_\beta}^T\,\Sigma_\beta\,[\nabla_a Q(s,a)]_{a=\mu_\beta}}}, \tag{23}$$

which completes the proof. $\qquad\square$

## A.2  Proof of Proposition 3.2

**Proposition 3.2.** *By applying the first inequality of Lemma 3.1 to the constraint of* (9), *we can derive an optimization problem that lower bounds* (9)

$$\max_{\mu} \quad \mu^T \left[\nabla_a Q(s,a)\right]_{a=a_\beta}$$
$$s.t. \quad \max_i \left\{ -\frac{1}{2}(\mu - \mu_i)^T \Sigma_i^{-1}(\mu - \mu_i) - \frac{1}{2}\log \det(2\pi\Sigma_i) + \log \lambda_i \right\} \geq -\delta, \tag{24}$$

*and the closed-form solution to* (10) *is given by*

$$\mu_{lse}(\tau) = \arg\max_{\bar{\mu}_i(\delta)} \quad \bar{\mu}_i^T \left[\nabla_a Q(s,a)\right]_{a=\mu_i}, \quad s.t. \quad \delta = \frac{1}{2}\min_i \left\{\log \lambda_i \det(2\pi\Sigma_i)\right\} + \log \tau$$

$$where \quad \bar{\mu}_i(\delta) = \mu_i + \sqrt{\frac{2(\delta + \log \lambda_i) - \log \det(2\pi\Sigma_i)}{\left[\nabla_a Q(s,a)\right]_{a=\mu_i}^T \Sigma_i \left[\nabla_a Q(s,a)\right]_{a=\mu_i}}} \Sigma_i \left[\nabla_a Q(s,a)\right]_{a=\mu_i} \tag{25}$$

*Proof.* Recall that the Gaussian Mixture behavior policy is constructed by

$$\pi_\beta = \sum_{i=1}^{N} \lambda_i \mathcal{N}(\mu_i, \Sigma_i), \tag{26}$$

We first divide the optimization problem (24) into $N$ sub-problems, with each sub-problem $i$ given by

$$\max_{\mu} \quad \mu^T \left[\nabla_a Q(s,a)\right]_{a=a_\beta}$$
$$s.t. \quad -\frac{1}{2}(\mu - \mu_i)^T \Sigma_i^{-1}(\mu - \mu_i) - \frac{1}{2}\log \det(2\pi\Sigma_i) + \log \lambda_i \geq -\delta, \tag{27}$$

which is equivalent to solving problem (5) for each Gaussian component with an additional constant term $\log \lambda_i$, and thus has a *unique* closed-form solution.

Define the maximizer for each sub-problem $i$ as $\bar{\mu}_i(\delta)$, though $\bar{\mu}_i(\delta)$ does not always exist. Whenever $-\frac{1}{2}\log \det(2\pi\Sigma_i) + \log \lambda_i < -\delta$, there will be no $\mu$ satisfying the constraint as $\frac{1}{2}(\mu - \mu_i)^T \Sigma_i^{-1}(\mu - \mu_i)$ is always greater than 0. We thus set $\bar{\mu}_i(\delta)$ to be *None* in this case. Next, we will show that there does not exist any $\breve{\mu} \notin \{\bar{\mu}_i(\delta)|i = 1 \ldots N\}$, s.t., $\breve{\mu}$ is the maximizer of (24). We can show this by contradiction. Suppose there exists a $\breve{\mu} \notin \{\bar{\mu}_i(\delta)|i = 1 \ldots N\}$ maximizing (24), there exists at least one $j \in \{1, \ldots, N\}$ s.t.

$$-\frac{1}{2}(\breve{\mu} - \mu_j)^T \Sigma_j^{-1}(\breve{\mu} - \mu_j) - \frac{1}{2}\log \det(2\pi\Sigma_j) + \log \lambda_j \geq -\delta. \tag{28}$$

Since $\breve{\mu}$ is the maximizer of (24), it should also be maximizer of the sub-problem $j$. However, the maximizer for sub-problem $j$ is given by $\bar{\mu}_j(\delta) \neq \breve{\mu}$, contradicting with the fact that $\breve{\mu}$ is the maximizer of the sub-problem $j$. Therefore, the optimal solution to (24) has to be given by

$$\arg\max_{\bar{\mu}_i} \quad \bar{\mu}_i^T \left[\nabla_a Q(s,a)\right]_{a=a_\beta} \quad where \quad \bar{\mu}_i \in \{\bar{\mu}_i(\delta)|i = 1 \ldots N\} \tag{29}$$

To solve each sub-problem $i$, it is natural to set $a_\beta = \mu_i$, which reformulate the sub-problem $i$ as below

$$\max_{\mu} \quad \mu^T \left[\nabla_a Q(s,a)\right]_{a=\mu_i}$$
$$s.t. \quad \frac{1}{2}(\mu - \mu_i)^T \Sigma_i^{-1}(\mu - \mu_i) \leq \delta - \frac{1}{2}\log \det(2\pi\Sigma_i) + \log \lambda_i, \tag{30}$$

Note that problem (30) is also a QCLP similar to the problem (5). Therefore, we can derive its solution by following similar procedures as in Appendix A.1, resulting in

$$\bar{\mu}_i(\delta) = \mu_i + \sqrt{\frac{2(\delta + \log \lambda_i) - \log \det(2\pi\Sigma_i)}{\left[\nabla_a Q(s,a)\right]_{a=\mu_i}^T \Sigma_i \left[\nabla_a Q(s,a)\right]_{a=\mu_i}}} \Sigma_i \left[\nabla_a Q(s,a)\right]_{a=\mu_i}. \tag{31}$$

We complete the proof by further setting $\delta = \frac{1}{2}\min_i \left\{\log \lambda_i \det(2\pi\Sigma_i)\right\} + \log \tau$. $\qquad\square$

### A.3 Proof of Proposition 3.3

**Proposition 3.3.** *By applying the second inequality of Lemma 3.1 to the constraint of (9), we can derive an optimization problem that lower bounds (9)*

$$\max_{\mu} \quad \mu^T \left[\nabla_a Q(s,a)\right]_{a=a_\beta}$$

$$s.t. \quad \sum_{i=1}^{N} \lambda_i \left(-\frac{1}{2}\log\det(2\pi\Sigma_i) - \frac{1}{2}(\mu-\mu_i)^T\Sigma_i^{-1}(\mu-\mu_i)\right) \geq -\delta \tag{32}$$

*and the closed-form solution to (12) is given by*

$$\mu_{jensen}(\tau) = \bar{\mu} + \sqrt{\frac{2\log\tau - \sum_{i=1}^{N}\lambda_i\mu_i^T\Sigma_i^{-1}\mu_i + \bar{\mu}^T\overline{\Sigma}^{-1}\bar{\mu}}{[\nabla_a Q(s,a)]_{a=\bar{\mu}}^T\overline{\Sigma}[\nabla_a Q(s,a)]_{a=\bar{\mu}}}} \; \overline{\Sigma}[\nabla_a Q(s,a)]_{a=\bar{\mu}},$$

$$where \; \overline{\Sigma} = \left(\sum_{i=1}^{N}\lambda_i\Sigma_i^{-1}\right)^{-1}, \quad \bar{\mu} = \overline{\Sigma}\left(\sum_{i=1}^{N}\lambda_i\Sigma_i^{-1}\mu_i\right), \quad \delta = \log\tau + \frac{1}{2}\sum_{i=1}^{N}\lambda_i\log\det(2\pi\Sigma_i) \tag{33}$$

*Proof.* Note that problem (32) is also a QCLP. Before deciding the value of $a_\beta$, we first derive its Lagrangian with a general $a_\beta$ below

$$L = \mu^T\left[\nabla_a Q(s,a)\right]_{a=a_\beta} - \eta\left(\sum_{i=1}^{N}\lambda_i\left(\frac{1}{2}\log\det(2\pi\Sigma_i) + \frac{1}{2}(\mu-\mu_i)^T\Sigma_i^{-1}(\mu-\mu_i)\right) - \delta\right) \tag{34}$$

Taking the derivatives w.r.t $\mu$, we get

$$\nabla_\mu L = [\nabla_a Q(s,a)]_{a=a_\beta} - \eta\left(\sum_{i=1}^{N}\lambda_i\left(\Sigma_i^{-1}(\mu-\mu_i)\right)\right) \tag{35}$$

By setting $\nabla_\mu L = 0$, we get

$$\mu = \left(\sum_{i=1}^{N}\lambda_i\Sigma_i^{-1}\right)^{-1}\left(\sum_{i=1}^{N}\lambda_i\Sigma_i^{-1}\mu_i\right) + \frac{1}{\eta}\left(\sum_{i=1}^{N}\lambda_i\Sigma_i^{-1}\right)[\nabla_a Q(s,a)]_{a=a_\beta}$$

$$= \bar{\mu} + \frac{1}{\eta}\overline{\Sigma}[\nabla_a Q(s,a)]_{a=a_\beta}, \tag{36}$$

$$where \quad \overline{\Sigma} = \left(\sum_{i=1}^{N}\lambda_i\Sigma_i^{-1}\right)^{-1}, \quad \bar{\mu} = \overline{\Sigma}\left(\sum_{i=1}^{N}\lambda_i\Sigma_i^{-1}\mu_i\right),$$

Equation 36 shows that the final solution to the problem (32) will be a shift from the pseudo-mean $\bar{\mu}$. Therefore, setting $a_\beta = \bar{\mu}$ becomes a natural choice.

Furthermore, by satisfying the KKT conditions, we have $\eta > 0$ and

$$\sum_{i=1}^{N}\lambda_i(\mu-\mu_i)^T\Sigma_i^{-1}(\mu-\mu_i) = 2\delta - \sum_{i=1}^{N}\lambda_i\log\det(2\pi\Sigma_i) \tag{37}$$

Plugging (32) into (37) gives the equation below

$$\sum_{i=1}^{N}\lambda_i\left(\bar{\mu} + \frac{1}{\eta}\overline{\Sigma}[\nabla_a Q(s,a)]_{a=\bar{\mu}} - \mu_i\right)^T\Sigma_i^{-1}\left(\bar{\mu} + \frac{1}{\eta}\overline{\Sigma}[\nabla_a Q(s,a)]_{a=\bar{\mu}} - \mu_i\right)$$

$$= 2\delta - \sum_{i=1}^{N}\lambda_i\log\det(2\pi\Sigma_i). \tag{38}$$

The LHS of (38) can be reformulated as

$$
\sum_{i=1}^{N} \lambda_i \left( \bar{\mu} + \frac{1}{\eta} \overline{\Sigma} \left[ \nabla_a Q(s,a) \right]_{a=\bar{\mu}} - \mu_i \right)^T \Sigma_i^{-1} \left( \bar{\mu} + \frac{1}{\eta} \overline{\Sigma} \left[ \nabla_a Q(s,a) \right]_{a=\bar{\mu}} - \mu_i \right)
$$

$$
= \frac{1}{\eta^2} \sum_{i=1}^{N} \lambda_i \left( \overline{\Sigma} \left[ \nabla_a Q(s,a) \right]_{a=\bar{\mu}} \right)^T \Sigma_i^{-1} \left( \overline{\Sigma} \left[ \nabla_a Q(s,a) \right]_{a=\bar{\mu}} \right)
$$

$$
+ \frac{2}{\eta} \sum_{i=1}^{N} \lambda_i \left( \overline{\Sigma} \left[ \nabla_a Q(s,a) \right]_{a=\bar{\mu}} \right)^T \Sigma_i^{-1} \left( \bar{\mu} - \mu_i \right)
$$

$$
+ \sum_{i=1}^{N} \lambda_i \left( \bar{\mu} - \mu_i \right)^T \Sigma_i^{-1} \left( \bar{\mu} - \mu_i \right)
$$

. (39)

We note that the second line of (39)'s RHS can be reduced to

$$
\frac{2}{\eta} \sum_{i=1}^{N} \lambda_i \left( \overline{\Sigma} \left[ \nabla_a Q(s,a) \right]_{a=\bar{\mu}} \right)^T \Sigma_i^{-1} \left( \bar{\mu} - \mu_i \right)
$$

$$
= \frac{2}{\eta} \left( \overline{\Sigma} \left[ \nabla_a Q(s,a) \right]_{a=\bar{\mu}} \right)^T \left( \left( \sum_{i=1}^{N} \lambda_i \Sigma_i^{-1} \right) \bar{\mu} - \sum_{i=1}^{N} \lambda_i \Sigma_i^{-1} \mu_i \right)
$$

$$
= \frac{2}{\eta} \left( \overline{\Sigma} \left[ \nabla_a Q(s,a) \right]_{a=\bar{\mu}} \right)^T \left( \overline{\Sigma}^{-1} \bar{\mu} - \overline{\Sigma}^{-1} \left( \overline{\Sigma} \sum_{i=1}^{N} \lambda_i \Sigma_i^{-1} \mu_i \right) \right)
$$

. (40)

$$
= \frac{2}{\eta} \left( \overline{\Sigma} \left[ \nabla_a Q(s,a) \right]_{a=\bar{\mu}} \right)^T \left( \overline{\Sigma}^{-1} \bar{\mu} - \overline{\Sigma}^{-1} \bar{\mu} \right)
$$

$$
= 0
$$

Therefore, (39) can be further reformulated as

$$
\sum_{i=1}^{N} \lambda_i \left( \bar{\mu} + \frac{1}{\eta} \overline{\Sigma} \left[ \nabla_a Q(s,a) \right]_{a=\bar{\mu}} - \mu_i \right)^T \Sigma_i^{-1} \left( \bar{\mu} + \frac{1}{\eta} \overline{\Sigma} \left[ \nabla_a Q(s,a) \right]_{a=\bar{\mu}} - \mu_i \right)
$$

$$
= \frac{1}{\eta^2} \sum_{i=1}^{N} \lambda_i \left( \overline{\Sigma} \left[ \nabla_a Q(s,a) \right]_{a=\bar{\mu}} \right)^T \Sigma_i^{-1} \left( \overline{\Sigma} \left[ \nabla_a Q(s,a) \right]_{a=\bar{\mu}} \right)
$$

$$
+ \sum_{i=1}^{N} \lambda_i \left( \bar{\mu} - \mu_i \right)^T \Sigma_i^{-1} \left( \bar{\mu} - \mu_i \right)
$$

$$
= \frac{1}{\eta^2} \left( \overline{\Sigma} \left[ \nabla_a Q(s,a) \right]_{a=\bar{\mu}} \right)^T \left( \sum_{i=1}^{N} \lambda_i \Sigma_i^{-1} \right) \left( \overline{\Sigma} \left[ \nabla_a Q(s,a) \right]_{a=\bar{\mu}} \right)
$$

. (41)

$$
+ \sum_{i=1}^{N} \lambda_i \left( \bar{\mu} - \mu_i \right)^T \Sigma_i^{-1} \left( \bar{\mu} - \mu_i \right)
$$

$$
= \frac{1}{\eta^2} \left( \overline{\Sigma} \left[ \nabla_a Q(s,a) \right]_{a=\bar{\mu}} \right)^T \overline{\Sigma}^{-1} \left( \overline{\Sigma} \left[ \nabla_a Q(s,a) \right]_{a=\bar{\mu}} \right)
$$

$$
+ \sum_{i=1}^{N} \lambda_i \left( \bar{\mu} - \mu_i \right)^T \Sigma_i^{-1} \left( \bar{\mu} - \mu_i \right)
$$

$$
= \frac{1}{\eta^2} \left[ \nabla_a Q(s,a) \right]_{a=\bar{\mu}}^T \overline{\Sigma} \left[ \nabla_a Q(s,a) \right]_{a=\bar{\mu}} + \sum_{i=1}^{N} \lambda_i \left( \bar{\mu} - \mu_i \right)^T \Sigma_i^{-1} \left( \bar{\mu} - \mu_i \right)
$$

To this point, (38) can be reformulated as

$$\frac{1}{\eta^2} \left[\nabla_a Q(s,a)\right]^T_{a=\bar{\mu}} \overline{\Sigma} \left[\nabla_a Q(s,a)\right]_{a=\bar{\mu}} + \sum_{i=1}^{N} \lambda_i \left(\bar{\mu} - \mu_i\right)^T \Sigma_i^{-1} \left(\bar{\mu} - \mu_i\right)$$

$$= 2\delta - \sum_{i=1}^{N} \lambda_i \log \det(2\pi\Sigma_i) \tag{42}$$

We can thus express $\eta$ as below

$$\eta = \sqrt{\frac{\left[\nabla_a Q(s,a)\right]^T_{a=\bar{\mu}} \overline{\Sigma} \left[\nabla_a Q(s,a)\right]_{a=\bar{\mu}}}{2\delta - \sum_{i=1}^{N} \lambda_i \log \det(2\pi\Sigma_i) - \sum_{i=1}^{N} \lambda_i \left(\bar{\mu} - \mu_i\right)^T \Sigma_i^{-1} \left(\bar{\mu} - \mu_i\right)}} \tag{43}$$

By setting $\delta = \frac{1}{2} \sum_{i=1}^{N} \lambda_i \log \det(2\pi\Sigma_i) + \log \tau$, we have

$$\eta = \sqrt{\frac{\left[\nabla_a Q(s,a)\right]^T_{a=\bar{\mu}} \overline{\Sigma} \left[\nabla_a Q(s,a)\right]_{a=\bar{\mu}}}{2 \log \tau - \sum_{i=1}^{N} \lambda_i \left(\bar{\mu} - \mu_i\right)^T \Sigma_i^{-1} \left(\bar{\mu} - \mu_i\right)}}$$

$$= \sqrt{\frac{\left[\nabla_a Q(s,a)\right]^T_{a=\bar{\mu}} \overline{\Sigma} \left[\nabla_a Q(s,a)\right]_{a=\bar{\mu}}}{2 \log \tau - \sum_{i=1}^{N} \lambda_i \bar{\mu}^T \Sigma_i^{-1} \bar{\mu} + 2\bar{\mu}^T \sum_{i=1}^{N} \lambda_i \Sigma_i^{-1} \mu_i - \sum_{i=1}^{N} \lambda_i \mu_i^T \Sigma_i^{-1} \mu_i}}$$

$$= \sqrt{\frac{\left[\nabla_a Q(s,a)\right]^T_{a=\bar{\mu}} \overline{\Sigma} \left[\nabla_a Q(s,a)\right]_{a=\bar{\mu}}}{2 \log \tau - \sum_{i=1}^{N} \bar{\mu}^T \overline{\Sigma}^{-1} \bar{\mu} + 2\bar{\mu}^T \overline{\Sigma}^{-1} \bar{\mu} - \sum_{i=1}^{N} \lambda_i \mu_i^T \Sigma_i^{-1} \mu_i}}$$

$$= \sqrt{\frac{\left[\nabla_a Q(s,a)\right]^T_{a=\bar{\mu}} \overline{\Sigma} \left[\nabla_a Q(s,a)\right]_{a=\bar{\mu}}}{2 \log \tau + \bar{\mu}^T \overline{\Sigma}^{-1} \bar{\mu} - \sum_{i=1}^{N} \lambda_i \mu_i^T \Sigma_i^{-1} \mu_i}} \tag{44}$$

Finally, plugging (44) into (36), with $a_\beta = \bar{\mu}$, we have

$$\mu_{\text{jensen}}(\tau) = \bar{\mu} + \sqrt{\frac{2 \log \tau - \sum_{i=1}^{N} \lambda_i \mu_i^T \Sigma_i^{-1} \mu_i + \bar{\mu}^T \overline{\Sigma}^{-1} \bar{\mu}}{\left[\nabla_a Q(s,a)\right]^T_{a=\bar{\mu}} \overline{\Sigma} \left[\nabla_a Q(s,a)\right]_{a=\bar{\mu}}}} \; \overline{\Sigma} \left[\nabla_a Q(s,a)\right]_{a=\bar{\mu}}, \tag{45}$$

which completes the proof. $\qquad\qquad\qquad\qquad\qquad\qquad\qquad\qquad\qquad\qquad\qquad\qquad\qquad\square$

## A.4 Proof of Theorem 3.1

In this section we prove the *safe policy improvement* presented in Section 3.3. Algorithm 1 follows the *approximate policy iteration* (API) (Perkins & Precup, 2002) by iterating over the policy evaluation ($\mathcal{E}$ step, Line 4) and policy improvement ($\mathcal{I}$ step, Line 5). Therefore, to verify $\mathcal{E}$ provides the improvement, we need to first show policy evaluation $\hat{Q}_t$ is accurate. In particular, we focus on the SARSA updates (Line 2), which is a form of on-policy Fitted Q-Iteration (Sutton & Barto, 2018). Fortunately, it is known that FQI is statistically efficient (*e.g.* (Chen & Jiang, 2019)) under the mild condition for the function approximation class. Its linear counterpart, least-square value iteration, is also shown to be efficient for offline reinforcement learning (Jin et al., 2021; Yin et al., 2022). Recently, (Zou et al., 2019) shows the finite sample convergence guarantee for SARSA under the standard the mean square error loss.

Next, to show the performance improvement, we leverage the performance difference lemma to show our algorithm achieves the desired goal.

**Lemma A.1** (Performance Difference Lemma). *For any policy $\pi, \pi'$, it holds that*

$$J(\pi) - J(\pi') = \frac{1}{1-\gamma}\mathbb{E}_{s\sim d^\pi}\left[\mathbb{E}_{a\sim\pi(\cdot|s)}A^{\pi'}(s,a)\right],$$

*where $A^\pi(s,a) = Q^\pi(s,a) - V^\pi(s)$ is the advantage function.*

Similar to (Kumar et al., 2020b), we focus on the discrete case where the number of states $|\mathcal{S}|$ and actions $|\mathcal{A}|$ are finite (note in the continuous case, the $\mathcal{D}(s,a)$ would be 0 for most locations, and thus the bound becomes less interesting). The adaptation to the continuous space can leverage standard techniques like *state abstraction* (Li et al., 2006) and covering arguments.

Next, we define the learning coefficient $C_{\gamma,\delta}$ of SARSA as

$$|\hat{Q}^{\hat{\pi}_\beta}(s,a) - Q^{\hat{\pi}_\beta}(s,a)| \leq \frac{C_{\gamma,\delta}}{\sqrt{\mathcal{D}(s,a)}}, \quad \forall s,a \in \mathcal{S} \times \mathcal{A}.$$

Define the first-order approximation error as

$$\bar{Q}^{\hat{\pi}_\beta}(s,a) := (a-a_\beta)^T\left[\nabla_a\hat{Q}^{\hat{\pi}_\beta}(s,a)\right]_{a=a_\beta} + \hat{Q}^{\hat{\pi}_\beta}(s,a_\beta),$$

then the approximation error is defined as:

$$C_{\text{CFPI}}(s,a) := |\bar{Q}^{\hat{\pi}_\beta}(s,a) - \hat{Q}^{\hat{\pi}_\beta}(s,a)| = \left|(a-a_\beta)^T\left[\nabla_a\hat{Q}^{\hat{\pi}_\beta}(s,a)\right]_{a=a_\beta} + \hat{Q}^{\hat{\pi}_\beta}(s,a_\beta) - \hat{Q}^{\hat{\pi}_\beta}(s,a)\right|.$$

Under the constraint $D\left(\pi(\cdot \mid s), \hat{\pi}_\beta(\cdot \mid s)\right) \leq \delta$ (4) (or equivalently action $a$ is close to $a_\beta$), the first-order approximation provides a good estimation for the $\hat{Q}^{\pi_\beta}$.

**Theorem A.1** (Restatement of Theorem 3.1). *Assume the state and action spaces are discrete. Let $\hat{\pi}_1$ be the policy obtained after the CFPI update (Line 2 of Algorithm 1). Then with probability $1 - \delta$,*

$$J(\hat{\pi}_1) - J(\hat{\pi}_\beta) \geq \frac{1}{1-\gamma}\mathbb{E}_{s\sim d^{\hat{\pi}_1}}\left[\bar{Q}^{\hat{\pi}_\beta}(s,\hat{\pi}_1(s)) - \bar{Q}^{\hat{\pi}_\beta}(s,\hat{\pi}_\beta(s))\right]$$

$$-\frac{2}{1-\gamma}\mathbb{E}_{s\sim d^{\hat{\pi}_1}}\mathbb{E}_{a\sim\hat{\pi}_1(\cdot|s)}\left[\frac{C_{\gamma,\delta}}{\sqrt{\mathcal{D}(s,a)}} + C_{\text{CFPI}}(s,a)\right] := \zeta.$$

*For multi-step $T$ iterative update, we similarly have with probability $1 - \delta$,*

$$J(\hat{\pi}_T) - J(\hat{\pi}_\beta) = \sum_{t=1}^T J(\hat{\pi}_t) - J(\hat{\pi}_{t-1}) \geq \sum_{t=1}^T \zeta^{(t)},$$

*where $\mathcal{D}(s,a)$ denotes number of samples at $s,a$, the learning coefficient of SARSA is defined as $C_{\gamma,\delta} = \max_{s_0,a_0}\sqrt{2\ln(12SA/\delta)} \cdot \sqrt{\sum_{h=0}^\infty\sum_{s,a}\gamma^{2h}\cdot\mu_h^{\hat{\pi}_\beta}(s,a|s_0,a_0)^2\text{Var}\left[V^{\hat{\pi}_\beta}(s')\mid s,a\right]}$, and $C_{CFPI}(s,a)$ denotes the error from the first-order approximation (3), (4) using CFPI, i.e. $C_{\text{CFPI}}(s,a) := \left|(a-a_\beta)^T\left[\nabla_a\hat{Q}^{\hat{\pi}_\beta}(s,a)\right]_{a=a_\beta} + \hat{Q}^{\hat{\pi}_\beta}(s,a_\beta) - \hat{Q}^{\hat{\pi}_\beta}(s,a)\right|$. When $a = a_\beta$, $C_{\text{CFPI}}(s,a) = 0$.*

*proof of Theorem 3.1.* We focus on the first update, which is from $\hat{\pi}_b$ to $\hat{\pi}_1$. According to the Sarsa update, we have $|\hat{Q}^{\hat{\pi}_\beta}(s,a) - Q^{\hat{\pi}_\beta}(s,a)| \leq \frac{C_{\gamma,\delta}}{\sqrt{\mathcal{D}(s,a)}}$, $\forall s,a \in \mathcal{S} \times \mathcal{A}$ with probability $1-\delta$ and this is due to previous on-policy evaluation result (*e.g.* (Zou et al., 2019)). Also denote $\hat{\pi}_1 := \arg\max_\pi \bar{Q}^{\hat{\pi}_\beta}$.

By Lemma A.1,

$$
\begin{aligned}
J(\hat{\pi}_1) - J(\hat{\pi}_\beta) =& \frac{1}{1-\gamma}\mathbb{E}_{s\sim d^{\hat{\pi}_1}}\left[\mathbb{E}_{a\sim\hat{\pi}_1(\cdot|s)}A^{\hat{\pi}_\beta}(s,a)\right] \\
=& \frac{1}{1-\gamma}\mathbb{E}_{s\sim d^{\hat{\pi}_1}}\left[\mathbb{E}_{a\sim\hat{\pi}_1(\cdot|s)}[Q^{\hat{\pi}_\beta}(s,a) - V^{\hat{\pi}_\beta}(s)]\right] \\
=& \frac{1}{1-\gamma}\mathbb{E}_{s\sim d^{\hat{\pi}_1}}\left[\mathbb{E}_{a\sim\hat{\pi}_1(\cdot|s)}[Q^{\hat{\pi}_\beta}(s,a) - Q^{\hat{\pi}_\beta}(s,\hat{\pi}_\beta(s))]\right] \\
\geq& \frac{1}{1-\gamma}\mathbb{E}_{s\sim d^{\hat{\pi}_1}}\left[\mathbb{E}_{a\sim\hat{\pi}_1(\cdot|s)}[\hat{Q}^{\hat{\pi}_\beta}(s,a) - \hat{Q}^{\hat{\pi}_\beta}(s,\hat{\pi}_\beta(s))]\right] - \frac{2}{1-\gamma}\mathbb{E}_{s\sim d^{\hat{\pi}_1}}\mathbb{E}_{a\sim\hat{\pi}_1(\cdot|s)}\left[\frac{C_{\gamma,\delta}}{\sqrt{\mathcal{D}(s,a)}}\right] \\
\geq& \frac{1}{1-\gamma}\mathbb{E}_{s\sim d^{\hat{\pi}_1}}\left[\bar{Q}^{\hat{\pi}_\beta}(s,\hat{\pi}_1(s)) - \bar{Q}^{\hat{\pi}_\beta}(s,\hat{\pi}_\beta(s))\right] - \frac{2}{1-\gamma}\mathbb{E}_{s\sim d^{\hat{\pi}_1}}\mathbb{E}_{a\sim\hat{\pi}_1(\cdot|s)}\left[\frac{C_{\gamma,\delta}}{\sqrt{\mathcal{D}(s,a)}} + C_{\text{CFPI}}(s,a)\right] \\
:=& \zeta^{(1)}
\end{aligned}
$$

where the first inequality uses $|\hat{Q}^{\hat{\pi}_\beta}(s,a) - Q^{\hat{\pi}_\beta}(s,a)| \leq \frac{C_{\gamma,\delta}}{\sqrt{\mathcal{D}(s,a)}}$ and the last inequality uses $\hat{\pi}_1 := \arg\max_\pi \bar{Q}^{\hat{\pi}_\beta}$. Here

$$
C_{\gamma,\delta} = \max_{s_0,a_0}\sqrt{2\ln(12SA/\delta)} \cdot \sqrt{\sum_{h=0}^\infty \sum_{s,a} \gamma^{2h} \cdot \mu_h^{\hat{\pi}_\beta}(s,a|s_0,a_0) \operatorname{Var}\left[V^{\hat{\pi}_\beta}(s') \mid s,a\right]}
$$

Similarly, if the number of iteration $t > 1$, then Denote

$$
C_{\gamma,\delta}^{(t)} := \max_{s_0,a_0}\sqrt{2\ln(12SA/\delta)} \cdot \sqrt{\sum_{h=0}^\infty \sum_{s,a} \gamma^{2h} \cdot \frac{\mu_h^{\hat{\pi}_t}(s,a|s_0,a_0)^2}{\mu_h^{\hat{\pi}_{t-1}}(s,a|s_0,a_0)} \operatorname{Var}\left[V^{\hat{\pi}_t}(s') \mid s,a\right]},
$$

then we have with probability $1-\delta$, by the Corollary 1 of Duan et al. (2020), the OPE estimation follows

$$
|\hat{Q}^{\hat{\pi}_\beta}(s,a) - Q^{\hat{\pi}_\beta}(s,a)| \leq \frac{C_{\gamma,\delta}^{(t)}}{\sqrt{\mathcal{D}(s,a)}}
$$

and

$$
\begin{aligned}
J(\hat{\pi}_t) - J(\hat{\pi}_{t-1}) \geq& \frac{1}{1-\gamma}\mathbb{E}_{s\sim d^{\hat{\pi}_t}}\left[\bar{Q}^{\hat{\pi}_{t-1}}(s,\hat{\pi}_t(s)) - \bar{Q}^{\hat{\pi}_{t-1}}(s,\hat{\pi}_{t-1}(s))\right] \\
& - \frac{2}{1-\gamma}\mathbb{E}_{s\sim d^{\hat{\pi}_t}}\mathbb{E}_{a\sim\hat{\pi}_t(\cdot|s)}\left[\frac{C_{\gamma,\delta}^{(t)}}{\sqrt{\mathcal{D}(s,a)}} + C_{\text{CFPI}}(s,a)\right] := \zeta^{(t)},
\end{aligned}
$$

then for multi-step iterative algorithm, by a union bound, we have with probability $1-\delta$

$$
J(\hat{\pi}_T) - J(\hat{\pi}_\beta) = \sum_{t=1}^T J(\hat{\pi}_t) - J(\hat{\pi}_{t-1}) \geq \sum_{t=1}^T \zeta^{(t)}.
$$

$\square$

**On the learning coefficient of SARSA.** The learning of SARSA is known to be statistically efficient from existing off-policy evaluation (OPE) literature, for instance (Duan et al., 2020; Yin & Wang, 2020). This is due to the on-policy SARSA scheme is just a special case of OPE task by choosing $\pi = \hat{\pi}_\beta$.

Concretely, we can translate the finite sample error bound in Corollary 1 of (Duan et al., 2020) to the infinite horizon discounted setting as: for any initial state,action $s_0, a_0$, with probability $1 - \delta$,

$$|\hat{Q}^{\hat{\pi}_\beta}(s_0, a_0) - Q^{\hat{\pi}_\beta}(s_0, a_0)| \leq \frac{1}{\sqrt{\mathcal{D}(s_0, a_0)}} \sqrt{2 \ln(12/\delta)} \cdot \sqrt{\sum_{h=0}^{\infty} \sum_{s,a} \gamma^{2h} \cdot \mu_h^{\hat{\pi}_\beta}(s, a | s_0, a_0) \operatorname{Var}\left[V^{\hat{\pi}_\beta}(s') \mid s, a\right]}$$

Note the original statement in (Duan et al., 2020) is for $v^{\hat{\pi}_\beta} - \hat{v}^{\hat{\pi}_\beta}$, here we conduct the version for $\hat{Q}^{\hat{\pi}_\beta} - Q^{\hat{\pi}_\beta}$ instead and this can be readily obtained by fixing the initial state action $s_0, a_0$ for $v^\pi$. As a result, by a union bound (over $S$, $A$) it is valid to define

$$C_{\gamma,\delta} = \max_{s_0, a_0} \sqrt{2 \ln(12SA/\delta)} \cdot \sqrt{\sum_{h=0}^{\infty} \sum_{s,a} \gamma^{2h} \cdot \mu_h^{\hat{\pi}_\beta}(s, a | s_0, a_0) \operatorname{Var}\left[V^{\hat{\pi}_\beta}(s') \mid s, a\right]}$$

and this makes sure the statistical guarantee in Theorem 3.1 follows through.

Similarly, for the multi-step case, the OPE estimator hold with the corresponding coefficient

$$C_{\gamma,\delta}^{(t)} := \max_{s_0, a_0} \sqrt{2 \ln(12SA/\delta)} \cdot \sqrt{\sum_{h=0}^{\infty} \sum_{s,a} \gamma^{2h} \cdot \frac{\mu_h^{\hat{\pi}_t}(s, a | s_0, a_0)^2}{\mu_h^{\hat{\pi}_{t-1}}(s, a | s_0, a_0)} \operatorname{Var}\left[V^{\hat{\pi}_t}(s') \mid s, a\right]}.$$

Lastly, even the assumption on the state-action space to be finite is not essential for Theorem 3.1 since, for more general function approximations, recent literature for OPE (Zhang et al., 2022) shows SARSA update in Algorithm 1 is still statistically efficient.

# B  Detailed Procedures to obtain Equation 14

We first highlight that we set the HP $\delta$ differently for Proposition 3.2 and 3.3. With the same $\tau$, we generate the two different $\delta$ for the two different settings. Specifically,

$$\delta_{\text{lse}}(\tau) = \log \tau + \min_i \left\{ \frac{1}{2} \log \det(2\pi\Sigma_i) - \log \lambda_i \right\}, \quad \text{(Proposition 3.2)}$$

$$\delta_{\text{jensen}}(\tau) = \log \tau + \frac{1}{2} \sum_{i=1}^{N} \lambda_i \log \det(2\pi\Sigma_i), \quad \text{(Proposition 3.3)}$$
(46)

We next provide intuition for the design choices (46). Recall that the Gaussian Mixture behavior policy is constructed by

$$\pi_\beta = \sum_{i=1}^{N} \lambda_i \mathcal{N}(\mu_i, \Sigma_i).$$
(47)

With the mixture weights $\lambda_{i=1...N}$, we define the scaled probability $\breve{\pi}_i(a)$ of the $i$-th Gaussian component evaluated at $a$

$$\breve{\pi}_i(\mu_i) = \lambda_i \pi_i(a) = \lambda_i \det(2\pi\Sigma_i)^{-\frac{1}{2}} \exp\{-\frac{1}{2}(a - \mu_i)^T \Sigma_i^{-1}(a - \mu_i)\},$$
(48)

where $\pi_i(a) = \mathcal{N}(a; \mu_i, \Sigma_i)$ denotes the probability of the $i$-th Gaussian component evaluated at $a$. Therefore, we can have $\log \breve{\pi}_i(\mu_i) = \log \lambda_i - \frac{1}{2} \log \det(2\pi\Sigma_i)$, which implies that

$$\delta_{\text{lse}}(\tau) = \log \tau + \min_i \left\{ \frac{1}{2} \log \det(2\pi\Sigma_i) - \log \lambda_i \right\}$$

$$= - \left( \max_i \left\{ \log \lambda_i - \frac{1}{2} \log \det(2\pi\Sigma_i) \right\} - \log \tau \right).$$
(49)

$$= - \max_i \left\{ \log \frac{1}{\tau} \breve{\pi}_i(\mu_i) \right\}.$$

By setting $\delta_{\text{lse}}(\tau)$ in this way, $\bar{\mu}_j = \bar{\mu}_j(\delta_{\text{lse}}(\tau))$ will satisfy the following condition whenever $\bar{\mu}_j$ is a valid solution to the sub-problem $j$ (27) due to the KKT conditions, $\forall j \in \{1, \dots, N\}$.

$$-\frac{1}{2}(\bar{\mu}_j - \mu_j)^T \Sigma_j^{-1}(\bar{\mu}_j - \mu_j) - \frac{1}{2} \log \det(2\pi\Sigma_j) + \log \lambda_j = -\delta_{\text{lse}}(\tau)$$

$$\iff \quad \log \breve{\pi}_j(\bar{\mu}_j) = \max_i \left\{ \log \frac{1}{\tau} \breve{\pi}_i(\mu_i) \right\} \quad \iff \quad \breve{\pi}_j(\bar{\mu}_j) = \frac{1}{\tau} \max_i \{\breve{\pi}_i(\mu_i)\}$$
(50)

To elaborate the design of $\delta_{\text{jensen}}(\tau)$, we first recall that the constraint of problem (12) is given by

$$\sum_{i=1}^{N} \lambda_i \left( -\frac{1}{2} \log \det(2\pi\Sigma_i) - \frac{1}{2}(\mu - \mu_i)^T \Sigma_i^{-1}(\mu - \mu_i) \right) \geq -\delta_{\text{jensen}}(\tau).$$
(51)

Note that the LHS of (51) is a concave function w.r.t $\mu$. Thus, we can obtain its maximum by setting its derivatives (52) to zero

$$\nabla_\mu \left( \sum_{i=1}^{N} \lambda_i \left( -\frac{1}{2} \log \det(2\pi\Sigma_i) - \frac{1}{2}(\mu - \mu_i)^T \Sigma_i^{-1}(\mu - \mu_i) \right) \right)$$

$$= - \sum_{i=1}^{N} \lambda_i \Sigma_i^{-1}(\mu - \mu_i) = -\overline{\Sigma}^{-1}\mu + \overline{\Sigma}^{-1}\bar{\mu}$$
(52)

Interestingly, we can find that the solution is given by $\mu = \bar{\mu}$. Plugging $\mu = \bar{\mu}$ into the LHS of (51), we can obtain its maximum as below

$$-\frac{1}{2} \sum_{i=1}^{N} \lambda_i \log \det(2\pi\Sigma_i) - \frac{1}{2} \sum_{i=1}^{N} \lambda_i (\bar{\mu} - \mu_i)^T \Sigma_i^{-1}(\bar{\mu} - \mu_i)$$

$$\leq \sum_{i=1}^{N} \lambda_i \left( -\frac{1}{2} \log \det(2\pi\Sigma_i) \right) = \sum_{i=1}^{N} \lambda_i \log \pi_i(\mu_i)$$
(53)

The inequality holds as the covariance matrix $\Sigma_i$ is a positive semi-definite matrix for $i \in \{1 \ldots N\}$. Therefore, our choice of $\delta_{\text{jensen}}(\tau)$ can be interpreted as

$$\delta_{\text{jensen}}(\tau) = \log \tau + \frac{1}{2} \sum_{i=1}^{N} \lambda_i \log \det(2\pi\Sigma_i) = -\left(\sum_{i=1}^{N} \lambda_i \log \pi_i(\mu_i) - \log \tau\right) \tag{54}$$

**Algorithm 2** Iterative $\mathcal{I}_{\text{MG}}$

---

1: **Input**: Learned behavior policy $\hat{\pi}_\beta$, Q network parameters $\phi_1$, $\phi_2$, target Q network parameters $\phi_{\text{targ},1}$, $\phi_{\text{targ},2}$, dataset $\mathcal{D}$, parameter $\tau$
2: **repeat**
3:    Randomly sample a batch of transitions, $B = \{(s, a, r, s', d)\}$ from $\mathcal{D}$
4:    Compute target actions

$$a'(s') = \text{clip}\left(\mathcal{I}_{\text{MG}}(\hat{\pi}_\beta, \hat{Q}; \tau)(s') + \text{clip}(\epsilon, -c, c), a_{Low}, a_{High}\right),$$

   where $\hat{Q} = \min(Q_{\phi_1}, Q_{\phi_2})$, and $\epsilon \sim \mathcal{N}(0, \sigma)$

5:    Compute targets

$$y(r, s', d) = r + \gamma(1 - d) \min_{i=1,2} Q_{\phi_{\text{targ},i}}(s', a'(s'))$$

6:    Update Q-functions by one step of gradient descent using

$$\nabla_{\phi_i} \frac{1}{|B|} \sum_{(s,a,r,s',d) \in B} \left(Q_{\phi_i}(s, a) - y(r, s', d)\right)^2 \qquad \text{for } i = 1, 2$$

7:    Update target networks with

$$\phi_{\text{targ},i} \leftarrow \rho\phi_{\text{targ},i} + (1 - \rho)\phi_i \qquad \text{for } i = 1, 2$$

8: **until** convergence
9: **Output**: $\mathcal{I}_{\text{MG}}(\hat{\pi}_\beta, \hat{Q}; \tau)$

---

## C   Multi-step and iterative algorithms

By setting $T > 0$, we can derive multi-step and iterative algorithms. Thanks to the tractability of our CFPI operators $\mathcal{I}_{\text{SG}}$ and $\mathcal{I}_{\text{MG}}$, we can always perform the policy improvement step in-closed form. Therefore, there is no significant gap between  multi-step and iterative algorithms with our CFPI operators. One can differentiate our multi-step and iterative algorithms by whether an algorithm trains the policy evaluation step $\mathcal{E}(\hat{Q}_{t-1}, \hat{\pi}_t, \mathcal{D})$ to convergence or not.

As for the policy evaluation operator $\mathcal{E}$, the fitted Q evaluation (Ernst et al., 2005; Le et al., 2019; Fujimoto et al., 2022) with a target network (Mnih et al., 2015) has been demonstrated to be an effective and successful paradigm to perform policy evaluation (Kumar et al., 2019; Fujimoto & Gu, 2021; Haarnoja et al., 2018; Lillicrap et al., 2015; Fujimoto et al., 2018) in deep (offline) RL. When instantiating a multi-step or iterative algorithm from Algorithm 1, one can also consider the other policy evaluation operators by incorporating more optimization techniques.

In the rest of this section, we will instantiate an iterative algorithm with our CFPI operators performing the policy improvement step and evaluate its effectiveness on the challenging AntMaze domains.

### C.1   Iterative algorithm with our CFPI operators

In Sec. 5.1, we instantiate an iterative algorithm *Iterative $\mathcal{I}_{MG}$* with our CFPI operator $\mathcal{I}_{\text{MG}}$. Algorithm 2 presents the corresponding pseudo-codes that learn a set of Q-function networks for simplicity. Without loss of generality, we can easily generalize the algorithm to learn the action-value distribution $Z(s, a)$ as is defined in (57).

For each task, we learn a Gaussian Mixture behavior policy $\hat{\pi}_\beta$ with behavior cloning. Similar to Sec. 5.1, we employed the IQN (Dabney et al., 2018a) architecture to model the Q-value network for its better generalizability. As our CFPI operator $\mathcal{I}_{\text{MG}}$ returns a deterministic policy, we follow the TD3 (Fujimoto et al., 2018) to perform policy smoothing by adding noise to the action $a'(s')$ in Line 4. After convergence, Algorithm 2 outputs an improved policy $\mathcal{I}_{\text{MG}}(\hat{\pi}_\beta, \hat{Q}; \tau)$.

Table 5 compares our Iterative $\mathcal{I}_{\text{MG}}$ with SOTA algorithms on the AntMaze domain. The performance for all baseline methods is directly reported from the IQL paper (Kostrikov et al., 2021). Our method

Table 5: Comparison between our iterative algorithm and SOTA methods on the AntMaze domain of D4RL. We report the mean and standard deviation across 5 seeds for our methods. Our Iterative $\mathcal{I}_{\mathrm{MG}}$ outperforms all baselines on 5 out of 6 tasks and obtaining the best overall performance, demonstrating the effectiveness of our CFPI operator when instantiating an iterative algorithm. "u" stands for umaze, "m" stands for medium, "l" stands for large, "p" stands for play, and "d" stands for diverse.

| Dataset | BC | DT | Onestep RL | TD3+BC | CQL | IQL | Iterative $\mathcal{I}_{\mathrm{MG}}$ |
|---|---|---|---|---|---|---|---|
| antmaze-u-v0 | 54.6 | 59.2 | 64.3 | 78.6 | 74.0 | 87.5 | **90.2 ± 3.9** |
| antmaze-u-d-v0 | 45.6 | 49.3 | 60.7 | 71.4 | **84.0** | 62.2 | 58.6 ± 15.2 |
| antmaze-m-p-v0 | 0.0 | 0.0 | 0.3 | 10.6 | 61.2 | 71.2 | **75.2 ± 6.9** |
| antmaze-m-d-v0 | 0.0 | 0.7 | 0.0 | 3.0 | 53.7 | 70.0 | **72.2 ± 7.3** |
| antmaze-l-p-v0 | 0.0 | 0.0 | 0.0 | 0.2 | 15.8 | 39.6 | **51.4 ± 7.7** |
| antmaze-l-d-v0 | 0.0 | 1.0 | 0.0 | 0.0 | 14.9 | 47.5 | **52.4 ± 10.9** |
| Total | 100.2 | 112.2 | 125.3 | 163.8 | 303.6 | 378.0 | **400.0 ± 52.0** |

outperforms all baseline methods on on 5 out of 6 tasks and obtaining the best overall performance. The training curves are shown in Fig. 3 with the HP settings detailed in Table 6. We did not perform much HP tuning, and thus one should expect a performance improvement after conducting fine-grained HP tuning.

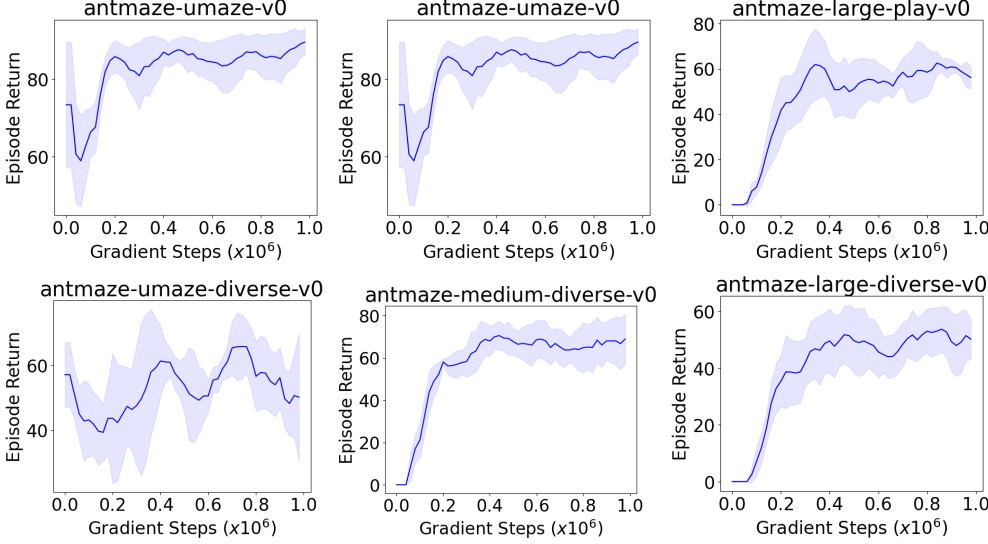

Figure 3: Iterative $\mathcal{I}_{\mathrm{MG}}$ training results on AntMaze. Shaded area denotes one standard deviation.

| | Hyperparameter | Value |
|---|---|---|
| Shared HP | Optimizer | Adam (Kingma & Ba, 2014) |
| | Normalize states | False |
| | activation function | ReLU |
| | Mini-batch size | 256 |
| MG-BC | Gaussian components ($N$) | 8 |
| | Number of gradient steps | 500K |
| | Policy architecture | MLP |
| | Policy learning rate | 1e-4 |
| | Policy hidden layers | 3 |
| | Policy hidden dim | 256 |
| | Threshold $\xi$ in (15) | 0.05 |
| Iterative $\mathcal{I}_{\mathrm{MG}}$ | Number of gradient steps | 1M |
| | Critic architecture | IQN (Dabney et al., 2018a) |
| | Critic hidden dim | 256 |
| | Critic hidden layers | 3 |
| | Critic learning rate | 3e-4 |
| | Number of quantiles $N_{\mathrm{q}}$ | 8 |
| | Number of cosine basis elements | 64 |
| | Discount factor | 0.99 |
| | Target update rate | 5e-3 |
| | Target update period | 1 |
| | $\log \tau$ | 1.5 |

Table 6: Hyperparameters for our Iterative $\mathcal{I}_{\mathrm{MG}}$.

## D   CFPI beyond Gaussian policies

In the main paper, we mainly discuss the scenario when the behavior policy $\pi_\beta$ is from the Gaussian family and develop two CFPI operators. However, our methods can also work with a non-Gaussian $\pi_\beta$. Next, we derive a new CFPI operator $\mathcal{I}_{\mathrm{DET}}$ that can work with deterministic $\pi_\beta$. We then show that $\mathcal{I}_{\mathrm{DET}}$ can also be leveraged to improve a general stochastic policy $\pi_\beta$ without knowing its actual expression, as long as we can sample from it.

### D.1   Deterministic behavior policy

When modeling both $\pi = \mu$ and $\pi_\beta = \mu_\beta$ as deterministic policies, we can derive the following BCPO from the problem (4) by setting $D(\cdot, \cdot)$ as the mean squared error.

$$\max_{\mu} \quad \mu^T \left[\nabla_a Q(s, a)\right]_{a=\mu_\beta}, \quad \text{s.t.} \quad \frac{1}{2}\|\mu - \mu_\beta\|^2 \leq \delta. \tag{55}$$

Problem (55) has a similar form as the problem (17). We can thus obtain its closed-form solution $\mu = \mu_{\mathrm{det}}(\delta)$ as below

$$\mu_{\mathrm{det}}(\delta) = \mu_\beta + \frac{\sqrt{2\delta}}{\|\left[\nabla_a Q(s, a)\right]_{a=\mu_\beta}\|} \left[\nabla_a Q(s, a)\right]_{a=\mu_\beta}. \tag{56}$$

Therefore, we can derive a new CFPI operator $\mathcal{I}_{\mathrm{DET}}(\pi_\beta, Q; \delta)$ that returns a policy with action selected by (56).

We further note that the problem (55) can be seen as a linear approximation of the objectives used in TD3 + BC (Fujimoto & Gu, 2021).

### D.2   Beyond deterministic behavior policy

Though we assume $\pi_\beta$ to be a deterministic policy during the derivation of $\mathcal{I}_{\mathrm{DET}}$, we can indeed leverage $\mathcal{I}_{\mathrm{DET}}$ to tackle the more general case when we can only sample from $\pi_\beta$ without knowing its actual expression.

**Algorithm 3** Policy improvement of $\mathcal{I}_{\text{DET}}$ with a stochastic $\pi_\beta$

---

**Input**: State $s$, stochastic policy $\pi_\beta$, value function $\hat{Q}$, $\delta$, number of candidate actions to sample $M$
 1: Sample candidate actions $\{a_1, \ldots, a_M\}$ from $\pi_\beta$
 2: Obtain the EBCQ policy $\pi_{\text{EBCQ}}$ with action selected by $\pi_{\text{EBCQ}}(s) = \arg\max_{m=1\ldots M} \hat{Q}(s, a_m)$
 3: Return $\mathcal{I}_{\text{DET}}(\pi_{\text{EBCQ}}, \hat{Q}; \delta)$ by calculating (56)

---

Table 7: $\mathcal{I}_{\text{DET}}$ results on the Gym-MuJoCo domain. We report the mean and standard deviation 5 seeds and each seed evaluates for 100 episodes.

| Dataset | DET-BC | VAE-BC | VAE-EBCQ | $\mathcal{I}_{\text{DET}}$ with $\pi_{\text{det}}$ | $\mathcal{I}_{\text{DET}}$ with $\pi_{\text{vae}}$ |
|---|---|---|---|---|---|
| Walker2d-Medium-v2 | $71.2 \pm 2.0$ | $70.6 \pm 3.0$ | $70.6 \pm 3.4$ | $79.5 \pm 12.9$ | $\mathbf{86.5 \pm 6.3}$ |
| Walker2d-Medium-Replay-v2 | $19.5 \pm 12.6$ | $19.4 \pm 2.9$ | $33.5 \pm 7.3$ | $57.1 \pm 11.6$ | $\mathbf{62.6 \pm 7.1}$ |
| Walker2d-Medium-Expert-v2 | $74.4 \pm 0.4$ | $74.9 \pm 7.6$ | $82.7 \pm 11.9$ | $\mathbf{111.2 \pm 1.8}$ | $\mathbf{111.1 \pm 0.9}$ |

Algorithm 3 details the procedures to perform the policy improvement step for a stochastic behavior policy $\pi_\beta$. We first obtain its EBCQ policy $\pi_{\text{EBCQ}}$ in Line 1-2. As $\pi_{\text{EBCQ}}$ is deterministic, we further plug it in $\mathcal{I}_{\text{DET}}$ in Line 3 to return an improved policy.

### D.3 Experiment results

To evaluate the performance of $\mathcal{I}_{\text{DET}}$, we first learn two behavior policies with two different models. Specifically, we model $\pi_{\text{det}}$ with a three-layer MLP that outputs a deterministic policy and $\pi_{\text{vae}}$ with the Variational auto-encoder (VAE) (Kingma & Welling, 2013) from BCQ (Fujimoto et al., 2019). Moreover, we reused the same value function $\hat{Q}_0$ as in Section 5.1. We present the results in Table 7. DET-BC and VAE denote the performance of $\pi_{\text{det}}$ and $\pi_{\text{vae}}$, respectively. VAE-EBCQ denotes the EBCQ performance of $\pi_{\text{vae}}$ with $M = 50$ candidate actions. Since $\pi_{\text{det}}$ is deterministic, its EBCQ performance is the same as DET-BC. As for our two methods, we set $\delta = 0.1$ for all datasets. We can observe that both our $\mathcal{I}_{\text{DET}}$ with $\pi_{\text{det}}$ and $\mathcal{I}_{\text{DET}}$ with $\pi_{\text{vae}}$ largely improve over the baseline methods. Moreover, $\mathcal{I}_{\text{DET}}$ with $\pi_{\text{vae}}$ outperforms VAE-EBCQ by a significant margins on all three datasets, demonstrating the effectiveness of our CFPI operator.

Indeed, our method benefits from an accurate and expressive behavior policy, as $\mathcal{I}_{\text{DET}}$ with $\pi_{\text{vae}}$ achieves a higher average performance compared to $\mathcal{I}_{\text{DET}}$ with $\pi_{\text{det}}$, while maintaining a lower standard deviation on all three datasets.

We also note that we did not spend too much effort optimizing the HP, e.g., the VAE architectures, learning rates, and the value of $\tau$.

# E    Reliable evaluation to address the statistical uncertainty

Figure 4: Comparison between our methods and baselines using reliable evaluation methods proposed in (Agarwal et al., 2021). We re-examine the results in Table 4 on the 9 tasks from the D4RL MuJoCo Gym domain. Each metric is calculated with a 95% CI bootstrap based on 9 tasks and 10 seeds for each task. Each seed further evaluates each method for 100 episodes. The interquartile mean (IQM) discards the top and bottom 25% data points and calculates the mean across the remaining 50% runs. The IQM is more robust as an estimator to outliers than the mean while maintaining less variance than the median. Higher median, IQM, mean scores, and lower Optimality Gap correspond to better performance. Our $\mathcal{I}_{MG}$ outperforms the baseline methods by a significant margin based on all four metrics.

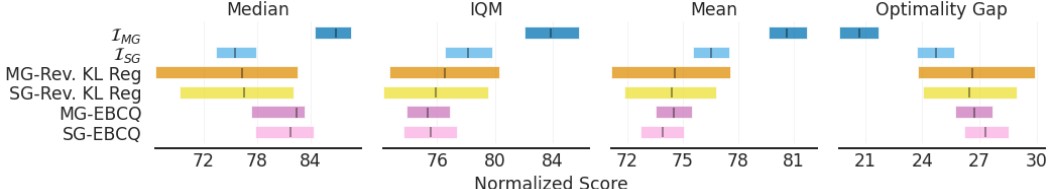

Figure 5: Performance profiles (score distributions) for all methods on the 9 tasks from the D4RL MuJoCo Gym domain. The average score is calculated by averaging all runs within one task. Each task contains 10 seeds, and each seed evaluates for 100 episodes. Shaded area denotes 95% confidence bands based on percentile bootstrap and stratified sampling (Agarwal et al., 2021). The $\eta$ value where the curves intersect with the dashed horizontal line $y = 0.5$ corresponds to the median, while the area under the performance curves corresponds to the mean.

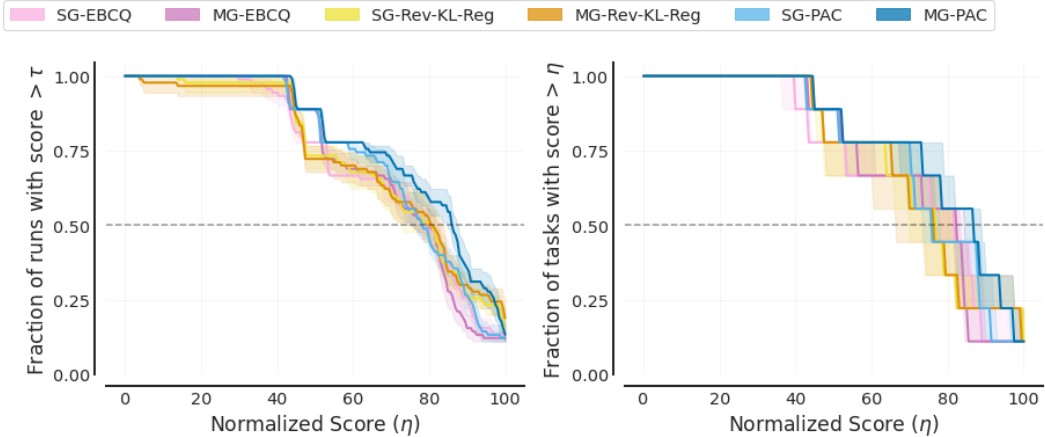

To demonstrate the superiority of our methods over the baselines and provide reliable evaluation results, we follow the evaluation protocols proposed in (Agarwal et al., 2021) to re-examine the results in Table 4. Specifically, we adopt the evaluation methods for all methods with $N_{\text{tasks}} \times N_{\text{seeds}}$ runs in total.

Moreover, we obtain the performance profile of each method, revealing its score distribution and variability. In particular, the score distribution shows the fraction of runs above a certain threshold $\eta$ and is given by

$$\hat{F}(\eta) = \hat{F}\left(\eta; x_{1:N_{\text{tasks}}, 1:N_{\text{seeds}}}\right) = \frac{1}{N_{\text{tasks}}} \sum_{m=1}^{N_{\text{tasks}}} \frac{1}{N_{\text{seeds}}} \sum_{n=1}^{N_{\text{seeds}}} \mathbb{1}\left[x_{m,n} \geq \eta\right]$$

Evaluation results in Fig. 4 and Fig. 5 demonstrate that our $\mathcal{I}_{MG}$ outperforms the baseline methods by a significant margin based on all four reliable metrics.

# F  Hyper-parameter settings and training details

For all methods we proposed in Table 1, Table 3, and Table 4, we obtain the mean and standard deviation of each method across 10 seeds. Each seed contains individual training process and evaluates the policy for 100 episodes.

## F.1  HP and training details for methods in Table 1 and Table 4

Table 8 includes the HP of methods evaluated on the Gym-MuJoCo domain. We use the Adam (Kingma & Ba, 2014) optimizer for all learning algorithms and normalize the states in each dataset following the practice of TD3+BC (Fujimoto & Gu, 2021). Note that our one-step offline RL algorithms presented in Table 1 (Our $\mathcal{I}_{\text{MG}}$) and Table 4 ($\mathcal{I}_{\text{MG}}$, $\mathcal{I}_{\text{SG}}$, MG-EBCQ, SG-EBCQ, MG-MS) require learning a behavior policy and the value function $\hat{Q}^0$. Therefore, we will first describe the detailed procedures for learning Single Gaussian (SG-BC) and Gaussian Mixture (MG-BC) behavior policies. We next describe our SARSA-style training procedures to estimate $\hat{Q}^0$. Finally, we will present the details for each one-step algorithm.

| | Hyperparameter | Value |
|---|---|---|
| Shared HP | Optimizer | Adam (Kingma & Ba, 2014) |
| | Normalize states | True |
| | Policy architecture | MLP |
| | Policy learning rate | 1e-4 |
| | Policy hidden layers | 3 |
| | Policy hidden dim | 256 |
| | Policy activation function | ReLU |
| | Threshold $\xi$ in (15) | 0.05 |
| MG-BC | Gaussian components ($N$) | 4 |
| | Number of gradient steps | 500K |
| | Mini-batch size | 256 |
| SG-BC | Number of gradient steps | 500K |
| | Mini-batch size | 512 |
| SARSA | Number of gradient steps | Table 16 |
| | Critic architecture | IQN (Dabney et al., 2018a) |
| | Critic hidden dim | 256 |
| | Critic hidden layers | 3 |
| | Critic activation function | ReLU |
| | Number of quantiles $N_{\text{q}}$ | 8 |
| | Number of cosine basis elements | 64 |
| | Discount factor | 0.99 |
| | Target update rate | 5e-3 |
| | Target update period | 1 |
| Our $\mathcal{I}_{\text{MG}}$ (Table 1) | $\log \tau$ | 0 for Hopper-M-E; 0.5 for the others |
| $\mathcal{I}_{\text{MG}}$ & $\mathcal{I}_{\text{SG}}$(Table 4) | $\log \tau$ | 0.5 for all tasks |
| MG-EBCQ | Number of candidate actions $N_{\text{bcq}}$ | 5 |
| SG-EBCQ | Number of candidate actions $N_{\text{bcq}}$ | 10 |
| MG-Rev. KL Reg | $\alpha$ | 3.0 |
| & SG-Rev. KL Reg | Number of gradient steps | 100K |

Table 8: Hyperparameters for our methods in Table 1 and Table 4.

**MG-BC.** We parameterize the policy as a 3-layer MLP, which outputs the tanh of a Gaussian Mixture with $N = 4$ Gaussian components. For each Gaussian component, we learn the state-dependent diagonal covariance matrix. While existing methods suggest learning Gaussian Mixture

via expectation maximization (Jordan & Jacobs, 1994; Xu et al., 1994; Jin et al., 2016) or variational Bayes (Bishop & Svensén, 2012), we empirically find that directly minimizing the negative log-likelihood of actions sampled from the offline datasets achieves satisfactory performance, as is shown in Table 1. We train the policy for 500K gradient steps. We emphasize that we do not aim to propose a better algorithm for learning a Gaussian Mixture behavior policy. Instead, future work may use a more advanced algorithm to capture the underlying behavior policy better.

**SG-BC.** We parameterize the policy as a 3-layer MLP, which outputs the tanh of a Single Gaussian with the state-dependent diagonal covariance matrix (Fu et al., 2020; Haarnoja et al., 2018). We train the policy for 500K gradient steps.

**SARSA.** We parameterize the value function with the IQN (Dabney et al., 2018a) architecture and train it to model the distribution $Z^\beta : \mathcal{S} \times \mathcal{A} \to \mathcal{Z}$ of the behavior return via quantile regression, where $\mathcal{Z}$ is the action-value distributional space (Ma et al., 2020) defined as

$$\mathcal{Z} = \{Z : \mathcal{S} \times \mathcal{A} \to \mathscr{P}(\mathbb{R}) \mid \mathbb{E}\left[|Z(s,a)|^p\right] < \infty, \forall (s,a), p \geq 1\}. \tag{57}$$

We define the CDF function of $Z^\beta$ as $F_{Z^\beta}(z) = Pr(Z^\beta < z)$, leading to the quantile function (Müller, 1997) $F_{Z^\beta}^{-1}(\rho) := \inf\{z \in \mathbb{R} : \rho \leq F_{Z^\beta}(z)\}$ as the inverse CDF function, where $\rho$ denotes the quantile fraction. We further denote $Z_\rho^\beta = F_{Z^\beta}^{-1}(\rho)$ to ease the notation.

To obtain $Z^\beta$, we leverage the empirical distributional bellman operator $\hat{\mathcal{T}}_D^\beta : \mathcal{Z} \to \mathcal{Z}$ defined as

$$\hat{\mathcal{T}}_D^\beta Z(s,a) :\overset{D}{=} r + \gamma Z(s',a') \mid (s,a,r,s',a') \sim \mathcal{D}, \tag{58}$$

where $A :\overset{D}{=} B$ implies the random variables $A$ and $B$ are governed by the same distribution. We note that $\hat{\mathcal{T}}_D^\beta$ helps to construct a Huber quantile regression loss (Dabney et al., 2018a; Ma et al., 2020; Dabney et al., 2018b), and we can finally learn $Z^\beta$ by minimizing the quantile regression loss following a similar procedures as in (Ma et al., 2020).

To achieve the goal, we approximate $Z^\beta$ by $N_q$ quantile fractions $\{\rho_i \in [0,1] \mid i = 0 \dots N_q\}$ with $\rho_0 = 0$, $\rho_{N_q} = 1$ and $\rho_i < \rho_j, \forall i < j$. We further denote $\hat{\rho}_i = (\rho_i + \rho_{i+1})/2$, and use random sampling (Dabney et al., 2018a) to generate the quantile fractions. By further parameterizing $Z_\rho^\beta(s,a)$ as $\hat{Z}_\rho^\beta(s,a;\theta)$ with parameter $\theta$, we can derive the loss function $J_Z(\theta)$ as

$$J_Z(\theta) = \mathbb{E}_{(s,a,r,s',a')\sim\mathcal{D}}\left[\sum_{i=0}^{N_q-1}\sum_{j=0}^{N_q-1}(\rho_{i+1} - \rho_i)\, l_{\hat{\rho}_j}(\delta_{ij})\right],$$

$$\text{where} \quad \delta_{ij} = \delta_{ij}(s,a,r,s',a') = r + \gamma Z_{\hat{\rho}_i}(s',a';\bar{\theta}) - Z_{\hat{\rho}_j}(s,a;\theta) \tag{59}$$

$$\text{and} \quad l_\rho(\delta_{ij}) = |\rho - \mathbb{I}\{\delta_{ij} < 0\}|\,\mathcal{L}(\delta_{ij}), \text{ with } \mathcal{L}(\delta_{ij}) = \begin{cases} \frac{1}{2}\delta_{ij}^2, & \text{if } |\delta_{ij}| \leq 1 \\ |\delta_{ij}| - \frac{1}{2}, & \text{otherwise.} \end{cases}$$

$\bar{\theta}$ is the parameter of the target network (Lillicrap et al., 2015) given by the Polyak averaging of $\theta$. We refer interested readers to (Dabney et al., 2018a; Ma et al., 2020) for further details.

The training procedures above returns $\hat{Z}_\rho^\beta, \forall \rho \in [0,1]$. With the learned $\hat{Z}_\rho^\beta$, our one-step methods presented in Table 1 and Table 4 extract the value function by setting $\hat{Q}_0 = \mathbb{E}_\rho[\hat{Z}_\rho^\beta] = \hat{Q}^\beta$ as the expectation of $\hat{Z}_\rho^\beta$, which is equivalent to the conventional action-value function $\hat{Q}^\beta$. Specifically, we use $N = 32$ fixed quantile fractions with $\rho_i = i/N, \ i = 0 \dots N$. Given a state-action pair $(s,a)$, we calculate $\hat{Q}_0(s,a) = \hat{Q}^\beta(s,a)$ as

$$\hat{Q}_0(s,a) = \hat{Q}^\beta(s,a) = \frac{1}{N}\sum_{i=1}^{N}\hat{Z}_{\hat{\rho}_i}^\beta(s,a), \quad \hat{\rho}_i = \frac{\rho_i + \rho_{i-1}}{2}. \tag{60}$$

Since our methods still need to query out-of-buffer action values during rollout, we employed the conventional double Q-learning (Fujimoto et al., 2018) technique to prevent potential overestimation without clipping. Specifically, we initialize $\hat{Q}_0^1$ and $\hat{Q}_0^2$ differently and train them to minimize (59). With the learned $\hat{Q}_0^1$ and $\hat{Q}_0^2$, we set the value of $\hat{Q}_0(s,a)$ as

$$\hat{Q}_0(s,a) = \min_{k=1,2}\hat{Q}_0^k(s,a) \tag{61}$$

Table 9: HP search for MG-EBCQ. We report the mean and std of 10 seeds, and each seed evaluates for 100 episodes.

| Dataset | $N_{\text{bcq}} = 2$ | $N_{\text{bcq}} = 5$ | $N_{\text{bcq}} = 10$ | $N_{\text{bcq}} = 20$ | $N_{\text{bcq}} = 50$ | $N_{\text{bcq}} = 100$ |
|---|---|---|---|---|---|---|
| Cheetah-M-v2 | $47.2 \pm 0.3$ | $51.5 \pm 0.2$ | $53.3 \pm 0.3$ | $54.4 \pm 0.3$ | $55.3 \pm 0.4$ | $55.8 \pm 0.4$ |
| Hopper-M-v2 | $63.3 \pm 2.3$ | $82.5 \pm 1.9$ | $88.3 \pm 4.6$ | $90.8 \pm 6.9$ | $92.1 \pm 7.6$ | $91.3 \pm 9.4$ |
| Walker2d-M-v2 | $78.6 \pm 1.4$ | $85.2 \pm 2.1$ | $81.0 \pm 6.3$ | $73.4 \pm 11.0$ | $67.6 \pm 14.8$ | $62.7 \pm 15.8$ |
| Cheetah-M-R-v2 | $38.8 \pm 0.6$ | $43.0 \pm 0.3$ | $44.2 \pm 0.3$ | $44.7 \pm 0.5$ | $44.8 \pm 0.8$ | $44.6 \pm 0.8$ |
| Hopper-M-R-v2 | $58.4 \pm 6.9$ | $83.6 \pm 10.3$ | $82.8 \pm 14.9$ | $82.3 \pm 15.9$ | $77.5 \pm 17.7$ | $76.0 \pm 16.7$ |
| Walker2d-M-R-v2 | $55.1 \pm 3.4$ | $73.1 \pm 5.2$ | $75.6 \pm 5.3$ | $77.6 \pm 5.4$ | $78.0 \pm 5.6$ | $78.5 \pm 4.5$ |
| Cheetah-M-E-v2 | $75.2 \pm 3.2$ | $84.5 \pm 4.6$ | $82.7 \pm 5.2$ | $77.6 \pm 7.3$ | $73.4 \pm 6.4$ | $68.8 \pm 5.9$ |
| Hopper-M-E-v2 | $73.6 \pm 7.5$ | $56.1 \pm 6.2$ | $44.9 \pm 4.6$ | $37.3 \pm 3.6$ | $29.8 \pm 2.9$ | $25.3 \pm 3.3$ |
| Walker2d-M-E-v2 | $107.1 \pm 1.8$ | $111.1 \pm 1.0$ | $111.4 \pm 1.5$ | $111.4 \pm 2.5$ | $109.6 \pm 4.0$ | $107.2 \pm 6.0$ |
| Total | $597.2 \pm 27.4$ | $670.6 \pm 31.9$ | $664.1 \pm 43.1$ | $649.5 \pm 53.5$ | $628.0 \pm 60.2$ | $610.2 \pm 62.9$ |

Table 10: HP search for SG-EBCQ. We report the mean and std of 10 seeds, and each seed evaluates for 100 episodes.

| Dataset | $N_{\text{bcq}} = 2$ | $N_{\text{bcq}} = 5$ | $N_{\text{bcq}} = 10$ | $N_{\text{bcq}} = 20$ | $N_{\text{bcq}} = 50$ | $N_{\text{bcq}} = 100$ |
|---|---|---|---|---|---|---|
| Cheetah-M-v2 | $47.1 \pm 0.2$ | $51.5 \pm 0.1$ | $53.3 \pm 0.2$ | $54.4 \pm 0.3$ | $55.3 \pm 0.3$ | $55.8 \pm 0.4$ |
| Hopper-M-v2 | $60.7 \pm 2.4$ | $78.6 \pm 4.0$ | $86.8 \pm 5.2$ | $89.1 \pm 7.7$ | $89.8 \pm 8.8$ | $89.8 \pm 9.8$ |
| Walker2d-M-v2 | $78.5 \pm 2.8$ | $86.9 \pm 1.8$ | $85.2 \pm 5.1$ | $81.5 \pm 9.3$ | $76.6 \pm 11.8$ | $72.4 \pm 13.8$ |
| Cheetah-M-R-v2 | $37.8 \pm 0.7$ | $42.3 \pm 0.6$ | $43.5 \pm 0.6$ | $44.3 \pm 0.7$ | $44.1 \pm 1.1$ | $43.6 \pm 0.9$ |
| Hopper-M-R-v2 | $58.7 \pm 5.8$ | $85.2 \pm 9.0$ | $88.5 \pm 12.2$ | $89.1 \pm 11.7$ | $83.9 \pm 15.0$ | $82.1 \pm 16.1$ |
| Walker2d-M-R-v2 | $54.0 \pm 7.2$ | $72.2 \pm 5.2$ | $75.4 \pm 4.6$ | $77.7 \pm 4.8$ | $77.5 \pm 5.8$ | $74.9 \pm 6.2$ |
| Cheetah-M-E-v2 | $71.8 \pm 2.2$ | $81.9 \pm 4.8$ | $81.8 \pm 5.4$ | $77.6 \pm 6.9$ | $71.5 \pm 7.5$ | $68.2 \pm 6.5$ |
| Hopper-M-E-v2 | $66.4 \pm 4.8$ | $49.8 \pm 6.2$ | $40.0 \pm 5.8$ | $34.9 \pm 6.2$ | $29.0 \pm 5.7$ | $25.2 \pm 4.8$ |
| Walker2d-M-E-v2 | $106.6 \pm 1.6$ | $111.0 \pm 0.9$ | $111.1 \pm 1.8$ | $110.0 \pm 3.7$ | $107.2 \pm 7.8$ | $106.0 \pm 9.0$ |
| Total | $581.6 \pm 27.7$ | $659.4 \pm 32.7$ | $665.5 \pm 41.0$ | $658.7 \pm 51.3$ | $634.7 \pm 63.9$ | $618.1 \pm 67.5$ |

for every $(s, a)$ pair. Note that the double Q-learning technique is only used during policy evaluation.

As for deciding the number of gradient steps, we detail our procedures in Appendix G.5. And the number of gradient steps for each dataset can be found in Table 16.

**Our $\mathcal{I}_{\text{MG}}$ (Table 1).** Recall that our CFPI operator $\mathcal{I}_{\text{MG}}(\hat{\pi}_\beta, \hat{Q}_0; \tau)$ requires to learn a Gaussian Mixture behavior policy $\hat{\pi}_\beta$ and a value function $\hat{Q}_0$. We train $\hat{\pi}_\beta$ and $\hat{Q}_0$ according to the procedures listed in **MG-BC** and **SARSA**, respectively. By following the practice of (Brandfonbrener et al., 2021; Fu et al., 2020), we perform a grid search on $\log \tau \in \{0, 0.5, 1.0, 1.5, 2.0\}$ using 3 seeds. We note that we manually reduce $\mathcal{I}_{\text{MG}}$ to MG-MS when $\log \tau = 0$ by only considering the mean of each non-trivial Gaussian component. Our results show that setting $\log \tau = 0.5$ achieves the best overall performance while Hopper-M-E requires an extremely small $\log \tau$ to perform well as is shown in Appendix G.2. Therefore, we decide to set $\log \tau = 0$ for Hopper-M-E and $\log \tau = 0.5$ for the other 8 datasets. We then obtain the results for the other 7 seeds with these HP settings and report the results on the 10 seeds in total.

**$\mathcal{I}_{\text{MG}}$ (Table 4) & $\mathcal{I}_{\text{SG}}$ (Table 4).** Different from the results in Table 1, we use the same $\log \tau = 0.5$ for all datasets including Hopper-M-E to obtain the performance of $\mathcal{I}_{\text{MG}}$ in Table 4. In this way, we aim to understand the effectiveness of each component of our methods better. To fairly compare $\mathcal{I}_{\text{MG}}$ and $\mathcal{I}_{\text{SG}}$, we tune the $\tau$ for $\mathcal{I}_{\text{SG}}$ in a similar way by performing a grid search on $\log \tau \in \{0.5, 1.0, 1.5, 2.0\}$ with 3 seeds and finally set $\log \tau = 0.5$ for all datasets. We then obtain the results for the other 7 seeds and report the results with 10 seeds in total.

**MG-EBCQ & SG-EBCQ.** We tune the number of candidate actions $N_{\text{bcq}}$ from the same range $\{2, 5, 10, 20, 50, 100\}$ as is in (Brandfonbrener et al., 2021). For each $N_{\text{bcq}}$, we obtain its average performance for all tasks across 10 seeds and select the best performing $N_{\text{bcq}}$ for each method. We separately tune the $N_{\text{bcq}}$ for MG-EBCQ and SG-EBCQ. As a result, we set $N_{\text{bcq}} = 5$ for MG-EBCQ and $N_{\text{bcq}} = 10$ for SG-EBCQ. **Moreover, we highlight that MG-EBCQ (SG-EBCQ) uses the same behavior policy and value function as is in $\mathcal{I}_{\text{MG}}$ ($\mathcal{I}_{\text{SG}}$).** We include the full hyper-parameter search results in Table 9 and Table 10.

Table 11: HP search for MG-Rev. KL Reg. We report the mean and std of 10 seeds, and each seed evaluates for 100 episodes.

| Dataset | $\alpha = 0.03$ | $\alpha = 0.1$ | $\alpha = 0.3$ | $\alpha = 1.0$ | $\alpha = 3.0$ | $\alpha = 10.0$ |
|---|---|---|---|---|---|---|
| Cheetah-M-v2 | $58.3 \pm 1.1$ | $58.1 \pm 1.2$ | $55.6 \pm 0.5$ | $50.6 \pm 0.3$ | $47.0 \pm 0.2$ | $44.5 \pm 0.2$ |
| Hopper-M-v2 | $14.4 \pm 13.6$ | $41.0 \pm 31.0$ | $89.4 \pm 22.2$ | $99.7 \pm 1.1$ | $76.3 \pm 6.9$ | $58.5 \pm 4.0$ |
| Walker2d-M-v2 | $5.8 \pm 4.8$ | $18.4 \pm 21.9$ | $34.2 \pm 27.3$ | $82.2 \pm 7.8$ | $82.8 \pm 1.8$ | $76.9 \pm 2.0$ |
| Cheetah-M-R-v2 | $46.7 \pm 1.8$ | $47.5 \pm 1.6$ | $48.1 \pm 0.7$ | $46.4 \pm 0.6$ | $44.4 \pm 0.5$ | $43.1 \pm 0.4$ |
| Hopper-M-R-v2 | $70.9 \pm 33.8$ | $86.6 \pm 26.3$ | $103.1 \pm 0.8$ | $101.4 \pm 1.1$ | $99.4 \pm 2.1$ | $77.6 \pm 17.2$ |
| Walker2d-M-R-v2 | $73.7 \pm 28.8$ | $65.4 \pm 33.8$ | $64.0 \pm 39.9$ | $65.4 \pm 35.8$ | $69.7 \pm 30.9$ | $57.7 \pm 22.8$ |
| Cheetah-M-E-v2 | $0.4 \pm 2.2$ | $1.2 \pm 1.9$ | $4.0 \pm 1.9$ | $25.0 \pm 6.3$ | $65.0 \pm 10.1$ | $86.2 \pm 7.1$ |
| Hopper-M-E-v2 | $2.6 \pm 1.7$ | $16.2 \pm 7.9$ | $22.5 \pm 10.7$ | $57.4 \pm 23.6$ | $79.4 \pm 32.6$ | $86.8 \pm 15.7$ |
| Walker2d-M-E-v2 | $10.4 \pm 15.3$ | $25.5 \pm 38.1$ | $93.5 \pm 34.5$ | $109.8 \pm 0.6$ | $107.1 \pm 4.0$ | $97.4 \pm 7.0$ |
| Total | $283.2 \pm 103.0$ | $359.9 \pm 163.5$ | $514.3 \pm 138.5$ | $637.8 \pm 77.2$ | $671.2 \pm 89.1$ | $628.6 \pm 76.4$ |

Table 12: HP search for SG-Rev. KL Reg. We report the mean and std of 10 seeds, and each seed evaluates for 100 episodes.

| Dataset | $\alpha = 0.03$ | $\alpha = 0.1$ | $\alpha = 0.3$ | $\alpha = 1.0$ | $\alpha = 3.0$ | $\alpha = 10.0$ |
|---|---|---|---|---|---|---|
| Cheetah-M-v2 | $58.6 \pm 1.3$ | $57.9 \pm 0.8$ | $55.2 \pm 0.5$ | $50.7 \pm 0.5$ | $47.1 \pm 0.2$ | $44.5 \pm 0.3$ |
| Hopper-M-v2 | $18.7 \pm 15.6$ | $40.2 \pm 24.7$ | $83.2 \pm 19.6$ | $98.8 \pm 2.0$ | $70.3 \pm 7.0$ | $57.2 \pm 4.6$ |
| Walker2d-M-v2 | $5.6 \pm 3.5$ | $26.2 \pm 27.1$ | $37.0 \pm 27.6$ | $83.3 \pm 7.5$ | $82.4 \pm 1.0$ | $77.1 \pm 1.2$ |
| Cheetah-M-R-v2 | $46.1 \pm 3.6$ | $47.8 \pm 1.3$ | $47.8 \pm 0.8$ | $46.0 \pm 0.5$ | $44.3 \pm 0.4$ | $42.5 \pm 0.6$ |
| Hopper-M-R-v2 | $77.4 \pm 19.1$ | $60.8 \pm 27.7$ | $92.0 \pm 21.9$ | $100.7 \pm 1.0$ | $99.7 \pm 1.0$ | $70.3 \pm 19.2$ |
| Walker2d-M-R-v2 | $59.5 \pm 31.3$ | $72.7 \pm 38.8$ | $75.7 \pm 30.4$ | $75.1 \pm 25.3$ | $63.6 \pm 28.5$ | $59.7 \pm 21.5$ |
| Cheetah-M-E-v2 | $1.1 \pm 3.2$ | $3.4 \pm 3.4$ | $7.1 \pm 4.3$ | $38.9 \pm 18.4$ | $78.9 \pm 9.8$ | $89.1 \pm 4.0$ |
| Hopper-M-E-v2 | $5.5 \pm 4.0$ | $20.1 \pm 8.6$ | $24.8 \pm 7.9$ | $43.8 \pm 23.6$ | $76.6 \pm 18.3$ | $67.7 \pm 30.6$ |
| Walker2d-M-E-v2 | $1.7 \pm 3.7$ | $13.4 \pm 33.5$ | $83.2 \pm 37.5$ | $109.9 \pm 0.7$ | $106.7 \pm 4.1$ | $96.8 \pm 7.6$ |
| Total | $274.0 \pm 85.3$ | $342.6 \pm 165.9$ | $505.9 \pm 150.5$ | $647.2 \pm 79.5$ | $669.7 \pm 70.3$ | $604.9 \pm 89.5$ |

**MG-Rev. KL Reg & SG-Rev. KL Reg.** We tune the regularization strength $\alpha$ from the same range $\{0.03, 0.1, 0.3, 1.0, 3.0, 10.0\}$ as is in (Brandfonbrener et al., 2021). For each $\alpha$, we obtain its average performance for all tasks across 10 seeds and select the best performing $\alpha$ for each method. We separately tune the $\alpha$ for MG-Rev. KL Reg & SG-Rev. KL Reg, although $\alpha = 3.0$ achieves the best overall performance in both methods. **Moreover, we highlight that MG-Rev. KL Reg (SG-Rev. KL Reg) uses the same behavior policy and value function as is in $\mathcal{I}_{MG}$ ($\mathcal{I}_{SG}$).** We include the full hyper-parameter search results in Table 11 and Table 12.

## F.2 HP and training details for methods in Table 3

| | Hyperparameter | Value |
|---|---|---|
| Shared HP | Normalize states | False |
| IQL HP | Optimizer | Adam (Kingma & Ba, 2014) |
| | Number of gradient steps | 1M |
| | Mini-batch size | 256 |
| | Policy learning rate | 3e-4 |
| | Policy hidden dim | 256 |
| | Policy hidden layers | 2 |
| | Policy activation function | ReLU |
| | Critic architecture | MLP |
| | Critic learning rate | 3e-4 |
| | Critic hidden dim | 256 |
| | Critic hidden layers | 2 |
| | Critic activation function | ReLU |
| | Target update rate | 5e-3 |
| | Target update period | 1 |
| | quantile | 0.9 |
| | temperature | 10.0 |
| $\mathcal{I}_{SG}(\pi_{IQL}, Q_{IQL})$ | $\log \tau$ | selected from $\{0.1, 0.2, 2.0\}$ |

Table 13: Hyperparameters for methods in Table 3

Table 13 includes the HP for experiments in Sec. 5.2. The of IQL. We use the same HP for the IQL training as is reported in the IQL paper. We obtain the IQL policy $\pi_{IQL}$ and $Q_{IQL}$ by training for 1M gradient steps using the PyTorch Implementation from RLkit (Berkeley), a widely used RL library. We emphasize that we follow the authors' exact training and evaluation protocol. We include the training curves for all tasks from the AntMaze domain in Appendix G.6.

Note that IQL (Kostrikov et al., 2021) reported inconsistent offline experiment results on AntMaze in its paper's Table 1, Table 2, Table 5, and Table 6 [3]. We suspect that these results are obtained from different sets of random seeds. In Appendix G.6, we present all these results in Table 17.

To obtain the performance for $\mathcal{I}_{SG}(\pi_{IQL}, Q_{IQL})$, we follow the practice of (Brandfonbrener et al., 2021; Fu et al., 2020) and perform a grid search on $\log \tau \in \{0.1, 0.2, 2.0\}$ using 3 seeds for each dataset. We then evaluate the best choice for each dataset by obtaining corresponding results on the other 7 seeds. We finally report the results with 10 seeds in total.

---

[3]Link to the IQL paper. IQL's Table 5 & 6 are presented in the supplementary material.

# G    Additional Experiments

## G.1    Complete experiment results for MG-MS

Table 14 provides the results of MG-MS on the 9 tasks from the MuJoCo Gym domain in compensation for the results in Sec. 5.3.

Table 14: Results of MG-MS on the MuJoCo Gym domain. We report the mean and standard deviation across 10 seeds, and each seed evaluates for 100 episodes.

| Dataset | MG-MS (15) |
|---------|------------|
| Cheetah-M-v2 | $43.6 \pm 0.2$ |
| Hopper-M-v2 | $55.3 \pm 6.3$ |
| Walker2d-M-v2 | $73.6 \pm 2.2$ |
| Cheetah-M-R-v2 | $42.4 \pm 0.4$ |
| Hopper-M-R-v2 | $61.5 \pm 15.1$ |
| Walker2d-M-R-v2 | $65.0 \pm 10.4$ |
| Cheetah-M-E-v2 | $91.3 \pm 2.1$ |
| Hopper-M-E-v2 | $104.2 \pm 5.1$ |
| Walker2d-M-E-v2 | $104.1 \pm 6.7$ |
| Total | $641.1 \pm 48.5$ |

## G.2  Complete experiment results on the effect of the HP $\tau$

Fig. 6 presents additional results in compensation for the results in Sec. 5.3. We note that Hopper-Medium-Expert-v2 requires a much smaller $\log \tau$ than the other tasks to perform well.

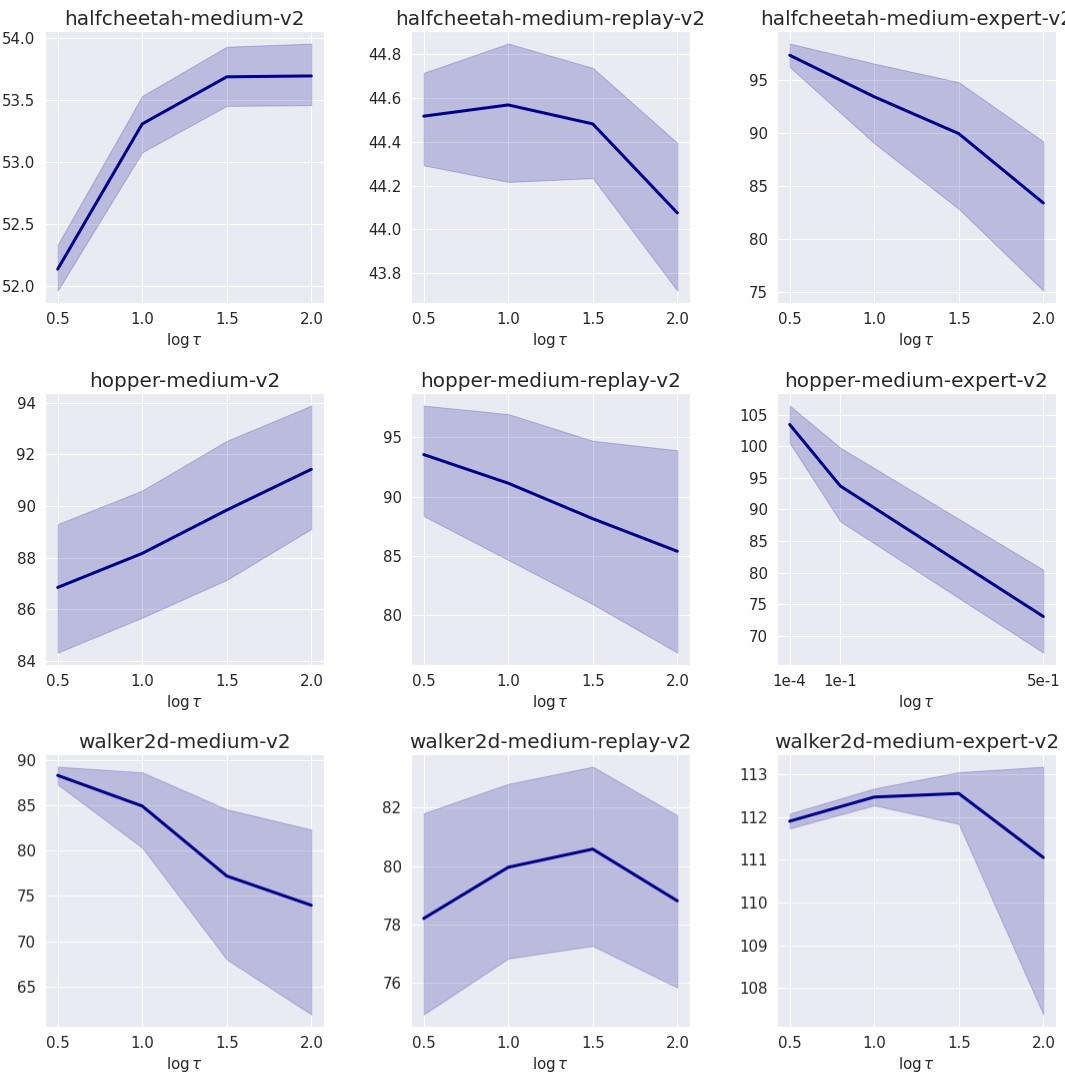

Figure 6: Performance of $\mathcal{I}_{\mathrm{MG}}$ with varying $\log \tau$. The other HP can be found in Table 8. Each variant averages returns over 10 seeds, and each seed contains 100 evaluation episodes. The shaded area denotes bootstrapped 95% CI.

## G.3 Ablation study on the number Gaussian components

In this section, we explore whether increasing the number of Gaussian components will result in a performance boost. We use the same settings as in Table 1 except modeling $\hat{\pi}_\beta$ with 8 Gaussian instead of 4. We hypothesize the performance gain should most likely happen on the three Medium-Replay datasets, as these datasets are collected by diverse policies. However, Table 15 shows that simply increasing the number of Gaussian components from 4 to 8 hardly results in a performance boost, as increasing the number of Gaussian components will induce extra optimization difficulties during behavior cloning (Jin et al., 2016).

Table 15: Comparison between setting the number of Gaussian components to 4 and 8 for our $\mathcal{I}_{\mathrm{MG}}$ on the three Medium-Replay datasets. We report the mean and standard deviation across 10 seeds, and each seed evaluates for 100 episodes.

| Dataset | 4 components (Table 1) | 8 components |
|---|---|---|
| Cheetah-M-R-v2 | $44.5 \pm 0.4$ | $44.3 \pm 0.3$ |
| Hopper-M-R-v2 | $93.6 \pm 7.9$ | $90.6 \pm 11.6$ |
| Walker2d-M-R-v2 | $78.2 \pm 5.6$ | $79.4 \pm 4.5$ |

## G.4 Modeling the value network with conventional MLP

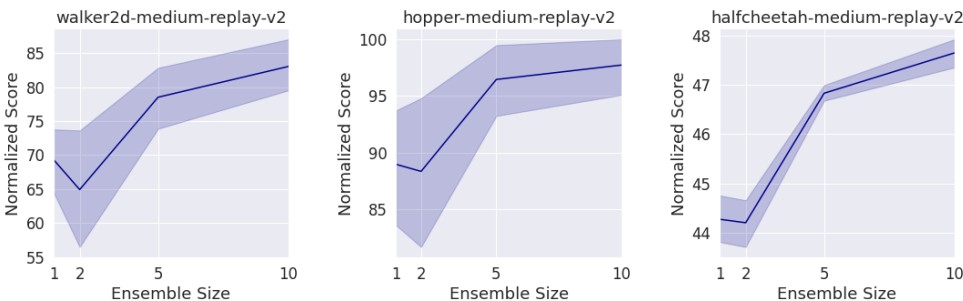

Figure 7: Performance of $\mathcal{I}_{\mathrm{MG}}$ with varying ensemble sizes. Each variant averages returns over 8 seeds, and each seed contains 100 evaluation episode. Each $Q$-value network is modeled by a 3-layer MLP. The shaded area denotes bootstrapped 95% CI.

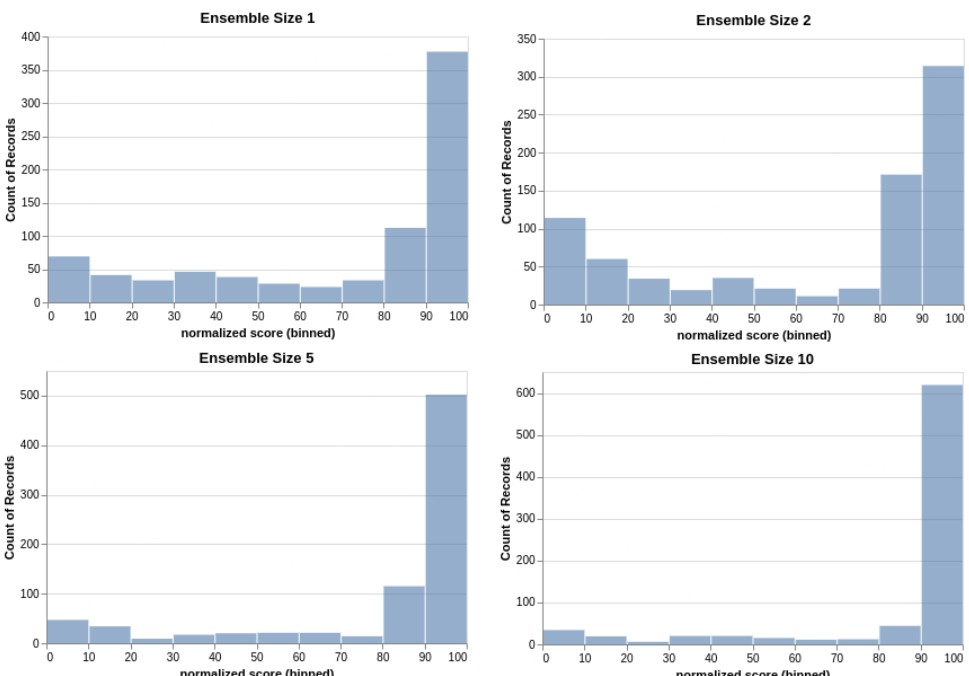

Figure 8: Performance of $\mathcal{I}_{\mathrm{MG}}$ with varying ensemble sizes on Walker2d-Medium-Replay-v2. Each variant aggregates returns over 8 seeds, and each seed evaluates for 100 episodes. Each $Q$-value network is modeled by a 3-layer MLP. With lower ensemble size, the performance exhibits large variance across different episodes.

Our experiments in Sec. 5.1 rely on learning a Q value function with the IQN (Dabney et al., 2018a) architecture. In this section, we examine the effectiveness of our CFPI operator $\mathcal{I}_{\mathrm{MG}}$ when working with an ensemble of conventional MLP $Q$-value networks with varying ensemble sizes $M$.

Each $Q$-value network $\hat{Q}^{\mathrm{MLP}}_{\theta_k}$ uses ReLU activation and is parameterized with $\theta_k$, including 3 hidden layers of width 256. We train each $\hat{Q}^{\mathrm{MLP}}_{\theta_k}$ by minimizing the bellman error below

$$L(\theta_k) = \mathbb{E}_{(s,a,r,s',a')\sim\mathcal{D}}\left[r + \gamma\hat{Q}^{\mathrm{MLP}}(s',a';\bar{\theta}_k) - \hat{Q}^{\mathrm{MLP}}(s,a;\theta_k)\right], \tag{62}$$

where $\bar{\theta}_k$ is the parameter of a target network given by the Polyak averaging of $\theta$. We set $\hat{Q}^{\text{MLP}}(s, a; \theta_k) = \hat{Q}^{\text{MLP}}_{\theta_k}(s, a)$. We further note that Equation 61 can be reformulated as

$$\hat{Q}_0(s, a) = \min_{k=1,2} \hat{Q}_0^k(s, a) = \frac{1}{2}|\hat{Q}_0^1(s, a) + \hat{Q}_0^2(s, a)| - \frac{1}{2}|\hat{Q}_0^1(s, a) - \hat{Q}_0^2(s, a)|$$
$$= \hat{\mu}_Q(s, a) - \hat{\sigma}_Q(s, a),$$
(63)

where $\hat{\mu}_Q$ and $\hat{\sigma}_Q$ calculate the mean and standard deviation of $Q$ value (Ciosek et al., 2019). In the case with an ensemble of $Q$, we obtain $\hat{Q}_0(s, a)$ by generalizing (63) as below

$$\hat{Q}_0(s, a) = \hat{\mu}_Q^{\text{MLP}} - \sqrt{\frac{1}{M} \sum_{k=1}^{M} \left( \hat{Q}^{\text{MLP}}(s, a; \theta_k) - \hat{\mu}_Q^{\text{MLP}} \right)^2},$$
(64)
$$\text{where} \quad \hat{\mu}_Q^{\text{MLP}} = \frac{1}{M} \sum_{k=1}^{M} \hat{Q}^{\text{MLP}}(s, a; \theta_k).$$

Other than the $Q$-value network, we applied the same setting as $\mathcal{I}_{\text{MG}}$ in Table 4. Fig. 7 presents the results with different ensemble sizes, showing that the performance generally increases with the ensemble size. Such a phenomenon illustrates a limitation of our CFPI operator $\mathcal{I}_{\text{MG}}$, as it heavily relies on accurate gradient information $\nabla_a[\hat{Q}_0(s, a)]_{a=a_\beta}$.

A large ensemble of $Q$ is more likely to provide accurate gradient information, thus leading to better performance. In contrast, a small ensemble size provides noisy gradient information, resulting in high variance across different rollout, as is shown in Fig. 8.

## G.5 How to decide the number of gradient steps for SARSA training?

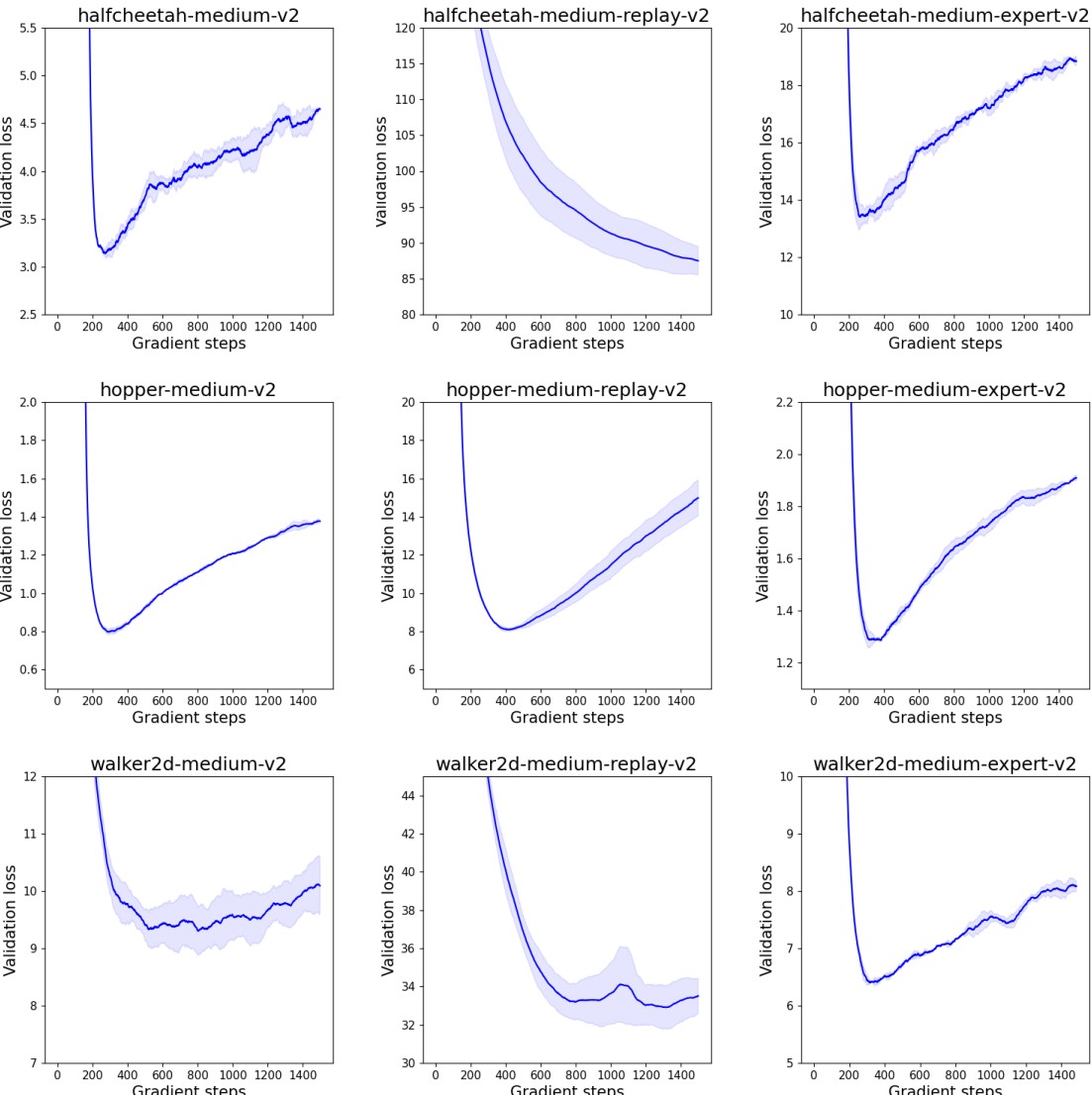

Figure 9: $\mathcal{L}_{\text{val}}$ on each dataset from the Gym-MuJoCo domain. We can observe that the model overfits to the training set when training for too may gradient steps. Each figure averages the validation loss over 2 folds with the same training seed. The shaded area denotes one standard deviation.

Deciding the number of gradient steps is a non-trivial problem in offline RL. While we use a fixed number of gradient steps for behavior cloning, we design a rigorous procedure to decide the gradient steps for SARSA training, inspired by the success of *k-fold validation*.

In our preliminary experiments, we first train a $\hat{Q}^{\beta}_{\text{all}}$ using all data from each dataset for 2M gradient steps. We model the $\hat{Q}^{\beta}_{\text{all}}(s, a)$ as a 3-layer MLP and train following Appendix G.4. By training in this way, we treat $\hat{Q}^{\beta}_{\text{all}}(s, a)$ as the ground truth $Q^{\beta}(s, a)$ for all $(s, a)$ sampled the dataset $\mathcal{D}$. Next, we randomly split the dataset with the ratio 95/5 to create the trainining set $\mathcal{D}_{\text{train}}$ validation set $\mathcal{D}_{\text{val}}$. We then train a new $\hat{Q}^{\beta}$ the SARSA training on $\mathcal{D}_{\text{train}}$. Therefore, we can define the validation loss as

$$\mathcal{L}_{\text{val}} = \mathbb{E}_{(s,a)\sim\mathcal{D}_{\text{val}}}||\hat{Q}^{\beta}_{\text{all}}(s, a) - \hat{Q}^{\beta}(s, a)||^2 \tag{65}$$

Fig. 9 presents the $\mathcal{L}_{\text{val}}$ on each dataset from the Gym-MuJoCo domain. We can clearly observe that $\hat{Q}^{\beta}$ generally overfits the $\mathcal{D}_{\text{train}}$ when training for too many gradient steps. We evaluate over two folds

Table 16: Gradient steps for the SARSA training

| Dataset | Gradient steps (K) |
|---|---|
| HalfCheetah-Medium-v2 | 200 |
| Hopper-Medium-v2 | 400 |
| Walker2d-Medium-v2 | 700 |
| HalfCheetah-Medium-Replay-v2 | 1500 |
| Hopper-Medium-Replay-v2 | 300 |
| Walker2d-Medium-Replay-v2 | 1100 |
| HalfCheetah-Medium-Expert-v2 | 400 |
| Hopper-Medium-Expert-v2 | 400 |
| Walker2d-Medium-Expert-v2 | 400 |

with one seed. Therefore, we can decide the gradient steps of each dataset for the SARSA training according to the results in Fig. 9 as listed in Table 16.

### G.6 Our reproduced IQL training curves

We use the PyTorch (Paszke et al., 2019) Implementation of IQL from RLkit (Berkeley) to obtain its policy $\pi_{\text{IQL}}$ and value function $Q_{\text{IQL}}$. We do not use the official implementation[4] open-sourced by the authors because our CFPI operators are also based on PyTorch. Fig. 10 presents our reproduced training curves of IQL on the 6 datasets from the AntMaze domain.

We note that the IQL paper[5] does not report consistent results in their paper for the offline experiment performance on the AntMaze, as is shown in Table 17. We suspect that these results are obtained from different sets of random seeds. Therefore, we can conclude that our reproduced results match the results reported in the IQL paper. We believe our reproduction results of IQL are reasonable, even if we do not use the official implementation open-sourced by the authors.

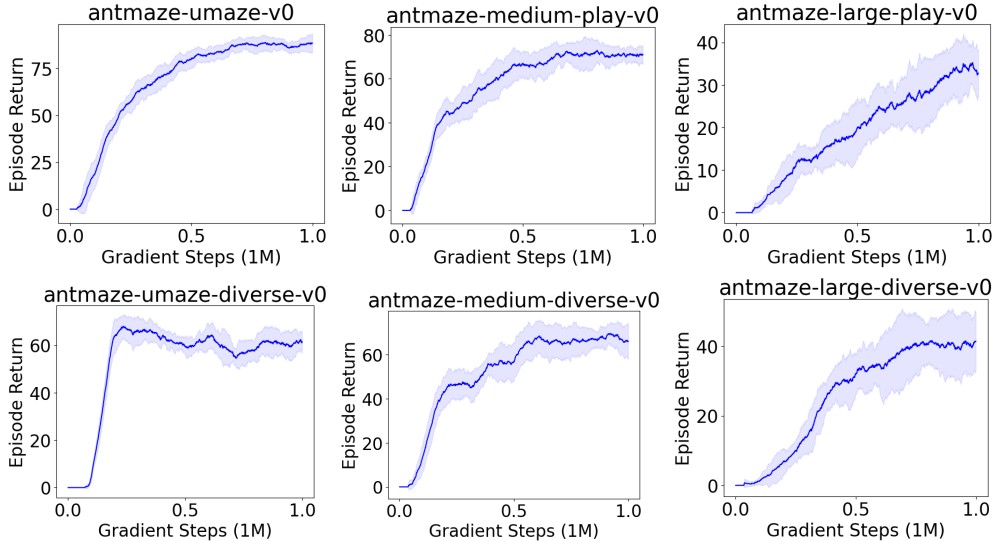

Figure 10: IQL offline training results on AntMaze. Shaded area denotes one standard deviation.

Table 17: Offline experiment results on AntMaze reported in different tables from the IQL paper

| Dataset | Table 1 & 6 | Table 2 | Table 5 |
|---|---|---|---|
| antmaze-u-v0 | $87.5 \pm 2.6$ | 88.0 | 86.7 |
| antmaze-u-d-v0 | $62.2 \pm 13.8$ | 67.0 | 75.0 |
| antmaze-m-p-v0 | $71.2 \pm 7.3$ | 69.0 | 72.0 |
| antmaze-m-d-v0 | $70.0 \pm 10.9$ | 71.8 | 68.3 |
| antmaze-l-p-v0 | $39.6 \pm 5.8$ | 36.8 | 25.5 |
| antmaze-l-d-v0 | $47.5 \pm 9.5$ | 42.2 | 42.6 |
| Total | $378.0 \pm 49.9$ | 374.8 | 370.1 |

---

[4] https://github.com/ikostrikov/implicit_q_learning
[5] Link to the IQL paper. IQL's Table 5 & 6 are presented in the supplementary material.

## G.7 Improve the policy learned by CQL

In this section, we show that our CFPI operators can also improve the policy learned by CQL (Kumar et al., 2020b) on the MuJoCo Gym Domain. We first obtain the CQL policy $\pi_{\text{CQL}}$ and $Q_{\text{CQL}}$ by training for 1M gradient steps using the official CQL implementation[6]. We obtain an improved policy $\mathcal{I}_{\text{SG}}(\pi_{\text{CQL}}, Q_{\text{CQL}}; \tau)$ that slightly outperforms $\pi_{\text{CQL}}$ overall, as shown in Table 18. For all 6 tasks, we set $\log \tau = 0.1$.

Table 18: Improving the policy learned by IQL with our CFPI operator $\mathcal{I}_{\text{SG}}$

| Dataset | $\pi_{\text{CQL}}$ (1M) | $\mathcal{I}_{\text{SG}}(\pi_{\text{CQL}}, Q_{\text{CQL}})$ |
|---|---|---|
| HalfCheetah-Medium-v2 | $45.5 \pm 0.3$ | $\mathbf{47.1 \pm 1.5}$ |
| Hopper-Medium-v2 | $65.4 \pm 3.5$ | $\mathbf{70.1 \pm 4.9}$ |
| Walker2d-Medium-v2 | $81.4 \pm 0.6$ | $\mathbf{81.6 \pm 1.1}$ |
| HalfCheetah-Medium-Replay-v2 | $44.6 \pm 0.5$ | $\mathbf{45.9 \pm 1.7}$ |
| Hopper-Medium-Replay-v2 | $95.2 \pm 2.0$ | $94.6 \pm 1.6$ |
| Walker2d-Medium-Replay-v2 | $80.1 \pm 2.6$ | $78.8 \pm 3.2$ |
| Total | $412.2 \pm 9.4$ | $\mathbf{418.2 \pm 13.9}$ |

---

[6]https://github.com/aviralkumar2907/CQL

# H    Additional related work

There has been a history of leveraging the Taylor expansion to construct efficient RL algorithms. Kakade & Langford (2002) proposed the conservative policy iteration that optimizes a mixture of policies towards its policy objective's lower bound, which is constructed by performing first-order Taylor expansion on the mixture coefficient. Later, SOTA deep RL algorithms TRPO (Schulman et al., 2015) and PPO (Schulman et al., 2017) extend the results to work with trust region policy constraints and learn a stochastic policy parameterized by a neural network. More recently, Tang et al. (2020) developed a second-order Taylor expansion approach under similar online RL settings.

At a high level, both our works and previous methods propose to create a surrogate of the original policy objective by leveraging the Taylor expansion approach. However, our motivation to use Taylor expansion is fundamentally different from the previous works (Kakade & Langford, 2002; Schulman et al., 2015, 2017; Tang et al., 2020), which leverage the Taylor expansion to construct a lower bound of the policy objective so that optimizing towards the lower bound translates into guaranteed policy improvement. However, these methods do not result in a closed-form solution to the policy and still require iterative policy updates.

On the other hand, our method leverages the Taylor expansion to construct a linear approximation of the policy objective, enabling the derivation of a closed-form solution to the policy improvement step and thus avoiding performing policy improvement via SGD. We highlight that our closed-form policy update cannot be possible without directly optimizing the parameter of the policy distribution. In particular, the parameter should belong to the action space. **We note that this is a significant conceptual difference between our method and previous works.**

Specifically, PDL (Kakade & Langford, 2002) parameterizes the mixture coefficient of a mixture policy as $\theta$. TRPO (Schulman et al., 2015) and PPO (Schulman et al., 2017) set $\theta$ as the parameter of a neural network that outputs the parameters of a Gaussian distribution. In contrast, our methods learn deterministic policy $\pi(s) = \text{Dirac}(\theta(s))$ and directly optimize the parameter $\theta(s)$. We aim to learn a greedy $\pi$ by solving $\theta(s) = \arg\max_a Q(s, a)$. However, obtaining a greedy $\pi$ in continuous control is problematic (Silver et al., 2014). Given the requirement of limited distribution shift in the offline RL, we thus leverage the first-order Taylor expansion to relax the problem into a more tractable form

$$\theta(s) = \arg\max_a \bar{Q}(s, a; a_\beta), \text{ s.t. } -\log \pi_\beta(a|s) \leq \delta, \tag{66}$$

where $\bar{Q}$ is defined in Equation 3. By modeling $\pi_\beta$ as a Single Gaussian or Gaussian Mixture, we further transform the problem into a QCLP and thus derive the closed-form solution.

Finally, we note that both the trust region methods TRPO and PPO and our methods constrain the divergence between the learned policy and behavior policy. However, the behavior policy always remains unchanged in our offline RL settings. As TRPO and PPO are designed for the online RL tasks, the updated policy will be used to collect new data and becomes the new behavior policy in future training iteration.

