# OpenReview forum: "Offline Reinforcement Learning with Closed-Form Policy Improvement Operators"
_NeurIPS.cc/2022/Workshop/Offline_RL — Offline RL Workshop NeurIPS 2022_

### Official Review · Reviewer_FPU7 · 2022-10-19

**Rating:** 6
**Confidence:** 4

**Review:**

This paper proposes closed-form policy improvement (CFPI) operator that avoid policy improvement step to learn a policy. This is motivated via the observation that the requirement of limited distributional shift can be naturally achieved via first-order Taylor approximation of the regularized RL objective. This lense admits closed form optimal solution when the behavior policy is Gaussian, and tractable optimization when the behavior policy is modeled as multivariate Gaussian. The practical implementation outperforms prior offline RL algorithms on the standard D4RL benchmark.

This is a sound paper with a simple idea. The performance improvement is marginal, though if the method is substantially more simpler, then this would be a good contribution to the literature. However, it is difficult to parse what the practical algorithm looks like from the main text.

---

### Official Review · Reviewer_FNdt · 2022-10-19

**Rating:** 6
**Confidence:** 4

**Review:**

This paper presents an offline RL algorithm that yields a closed-form policy improvement operator. This method leverages first-order approximation of the Q-function and achieves a linear approximation of the policy objective. The authors also propose to use a Gaussian Mixture for modeling the behavior policies with heterogeneous behaviors. Empirical results show that the method can outperform prior methods on D4RL benchmarks.

Pros:
1. Deriving a closed-form policy improvement operator is a neat idea for handling distributional shift.
2. The empirical results suggest that the method can outperform prior offline RL methods, which is pretty impressive.

Cons:
1. This method requires learning a Gaussian Mixture behavior policy. I wonder if the method will have trouble in scenarios where behavior policies are highly multimodal and hard to model.
2. The authors should show results in more complex domains such as Adroit, Antmaze or Kitchen. In those cases, using the first-order approximation of Q-functions might not work due to the complexity of the problem.